# Evidence for charge delocalization crossover in the quantum critical superconductor CeRhIn₅

Honghong Wang[1,2,5], Tae Beom Park[1,2,3,5], Jihyun Kim[1,2], Harim Jang[1,2], Eric D. Bauer [4], Joe D. Thompson[4] ✉ & Tuson Park [1,2] ✉

The nature of charge degrees-of-freedom distinguishes scenarios for interpreting the character of a second order magnetic transition at zero temperature, that is, a magnetic quantum critical point (QCP). Heavy-fermion systems are prototypes of this paradigm, and in those, the relevant question is where, relative to a magnetic QCP, does the Kondo effect delocalize their $f$-electron degrees-of-freedom. Herein, we use pressure-dependent Hall measurements to identify a finite-temperature scale $E_{loc}$ that signals a crossover from $f$-localized to $f$-delocalized character. As a function of pressure, $E_{loc}(P)$ extrapolates smoothly to zero temperature at the antiferromagnetic QCP of CeRhIn₅ where its Fermi surface reconstructs, hallmarks of Kondo-breakdown criticality that generates critical magnetic and charge fluctuations. In 4.4% Sn-doped CeRhIn₅, however, $E_{loc}(P)$ extrapolates into its magnetically ordered phase and is decoupled from the pressure-induced magnetic QCP, which implies a spin-density-wave (SDW) type of criticality that produces only critical fluctuations of the SDW order parameter. Our results demonstrate the importance of experimentally determining $E_{loc}$ to characterize quantum criticality and the associated consequences for understanding the pairing mechanism of superconductivity that reaches a maximum $T_c$ in both materials at their respective magnetic QCP.

The Kondo singlet is a quantum state in which spins of surrounding conduction electrons collectively screen a local moment through their antiferromagnetic (AFM) exchange. Theoretically expected and experimentally confirmed[1,2], the cloud of screening conduction electrons around a Kondo impurity extends radially to a distance $\xi = \hbar v_F / k_B T_K$, where $v_F$ is the conduction electron Fermi velocity and $k_B T_K$ is the energy scale of singlet formation. $\xi$ can be up to micrometers and certainly greater than interatomic spacing. In the many-body process of Kondo-singlet formation, the spin of the local moment becomes part of the conduction electron Fermi volume. For a periodic lattice of Kondo impurities, typified by $f$-electron heavy-fermion metals,

quantum coherence among Kondo singlets (qualitatively, a Bloch state of Kondo-screening clouds) results in the formation of highly entangled composite heavy quasiparticles as $T \to 0$ K and an increase in the Fermi volume that counts both the local moments and conduction electrons. Interactions within the narrow (of order meV) quasiparticle bands can lead to an instability, often a spin-density-wave (SDW), of the large Fermi volume. Tuning the SDW transition to $T = 0$ K by a nonthermal control parameter, such as pressure, magnetic field, or chemical substitution, allows access to a quantum critical point (QCP) in which quantum fluctuations of the SDW order parameter control physical properties to temperatures well above $T = 0$ K[3–5]. The

[1]Center for Quantum Materials and Superconductivity (CQMS), Sungkyunkwan University, Suwon, South Korea. [2]Department of Physics, Sungkyunkwan University, Suwon, South Korea. [3]Institute of Basic Science, Sungkyunkwan University, Suwon, South Korea. [4]Los Alamos National Laboratory, Los Alamos, NM, USA. [5]These authors contributed equally: Honghong Wang, Tae Beom Park. ✉e-mail: jdt@lanl.gov; tp8701@skku.edu

transition from a SDW order to a paramagnetic state at $T = 0$ K has no effect on the Fermi volume. This picture of a SDW QCP ignores the role of a long-range Rudermann-Kittel-Kasuya-Yosida (RKKY) interaction $I$ among local moments that is mediated by the same conduction electrons that produce a Kondo singlet. The RKKY interaction induces dynamical correlations among the local moments, inhibiting Kondo singlet formation and thus preventing the emergence of a large Fermi volume. The competition between Kondo and RKKY interactions can be characterized by a non-thermal tuning parameter $\delta = k_B T_K / I$[6,7]. Relative to the Kondo scale, $I$, though always finite, is relatively insensitive to pressure, field, etc. so that the $T-\delta$ phase diagram illustrated in Fig. 1 for heavy-fermion systems is controlled primarily by changes in $T_K$.

The nature of quantum critical fluctuations at $\delta_c$ where the magnetic boundary $T_N(\delta)$ reaches $T = 0$ K changes qualitatively depending on the location of a crossover scale $E_{loc}(\delta)$ that separates an electronic state in which static Kondo entanglement breaks down because of RKKY interactions and the Fermi surface is small ($FS_S$) and a state with fully intact composite quasiparticles that produce a large Fermi surface ($FS_L$). Depending on the position of $E_{loc}(\delta)$, the nature of the magnetic QCP at $\delta_c$ is either of the SDW or Kondo-breakdown type. In the SDW scenario (Fig. 1a), $E_{loc}$ remains finite at the QCP and terminates inside the ordered phase; only magnetic degrees-of-freedom are quantum critical at $\delta_c$[3]. In a Kondo-breakdown scenario (Fig. 1b), however, $E_{loc}$ reaches zero temperature at the magnetic QCP and the concomitant reconstruction of the FS from small-to-large coincides with the onset of magnetic order, thus incorporating both charge and magnetic quantum fluctuations at the Kondo-breakdown QCP[8-11]. The distinct difference between SDW and Kondo-breakdown criticality has fundamental consequences for interpreting the origin of new phases of matter that frequently emerge around a QCP as the system relieves the buildup of entropy.

Evidence for a Kondo-breakdown type of quantum criticality has been found in a few heavy-fermion materials, notably $CeCu_{6-x}Au_x$[12], $YbRh_2Si_2$[13,14], and $CeRhIn_5$[15,16] whereas, a phase diagram like that in Fig. 1a appears in $CeIn_3$[17], Co-doped $YbRh_2Si_2$[18] and Ir-doped $CeRhIn_5$[19,20]. Among examples representative of Fig. 1b physics, a FS change at their QCP has been inferred from Hall measurements. In $CeRhIn_5$, on the other hand, evidence for an abrupt change in FS as $T \to 0$ K has come from a pressure-dependent quantum oscillation study that directly probes the FS of $CeRhIn_5$ in which an abrupt reconstruction of the FS coincides with an AFM

QCP[15,21,22]. Evidence for the crossover scale $E_{loc}(\delta)$, characteristic of Kondo breakdown, has yet to be reported in $CeRhIn_5$, opening the possibility that FS reconstruction could be a consequence of a change in FS topology resulting from the loss of magnetic order[23]. Here, we probe the change in Hall effect via systematic control of external pressure to identify $E_{loc}(\delta)$ in $CeRhIn_5$, thereby demonstrating clearly that its criticality is of the Kondo-breakdown type. Replacing 4.4% of the In atoms by Sn shifts $E_{loc}(\delta)$ such that it intersects the $T_N(P)$ boundary at finite temperature (Fig. 1a) and the magnetic criticality changes to the SDW type. Not only imposing a far more unambiguous interpretation of criticality in $CeRhIn_5$, these discoveries point to a change in the nature of fluctuations leading to Cooper pairing in pure and Sn-doped $CeRhIn_5$ and to the importance of $E_{loc}$ for confirming a theoretically proposed beyond-Landau framework to understand quantum criticality[4,6].

## Results

### Observation of $E_{loc}$ in $CeRhIn_5$

Previous Hall measurements of $CeRhIn_5$ were limited to 2.6 GPa close to the critical pressure, $P_c = 2.3$ GPa, at which the AFM transition $T_N$ extrapolates to $T = 0$ K and the superconducting transition temperature $T_c$ reaches a maximum[21,22,24]. Our measurements of the Hall coefficient $R_H$ at $P < P_c$, Fig. 2a, are consistent with the primary features reported earlier[24]. A local minimum in $R_H$ at $T^*$ (as indicated by purple arrows) signals the onset of short-range AFM spin correlations above $T_N$[25,26] and is suppressed together with $T_N$ under pressure. Figure 2b shows $R_H$ in the high-pressure regime ($P > P_c$) where a previously unidentified local minimum in $R_H$ appears at $T_L$ (as marked by the orange arrows) and moves to higher temperature as the system is tuned away from magnetic order with increasing pressure. ($T_L$ appears well above the temperature below which the resistivity assumes a Fermi-liquid $T^2$ dependence[16].) The Hall coefficient in a multiband system with magnetic ions, particularly a compensated heavy-fermion metal like $CeRhIn_5$, is not straightforward to interpret[15,24,27]. Prior experiments rule out any significant asymmetric (skew) scattering in $CeRhIn_5$ at low temperatures and pressures below $P_c$[24,27], leaving temperature- and pressure-dependent changes in the ordinary Hall contribution as the most likely origin of $T_L$. In a system like $CeRhIn_5$, the magnitude of the ordinary term can be influenced not only by changes in carrier density but also by scattering rates on different parts of the Fermi surface, details of the surface topology, and the presence of spin fluctuations. Likely, all contribute to some extent to the unusual temperature- and

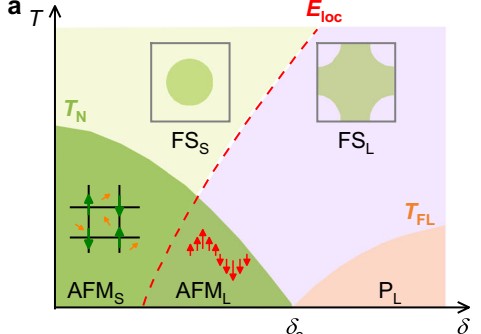

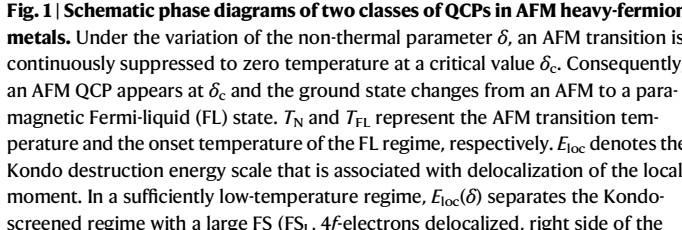

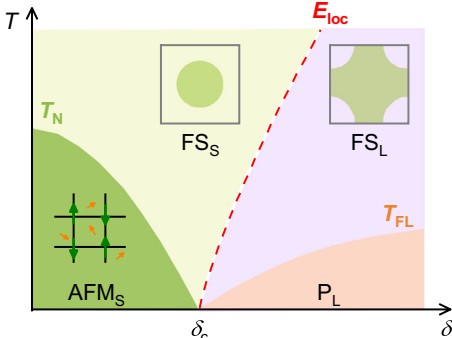

**Fig. 1 | Schematic phase diagrams of two classes of QCPs in AFM heavy-fermion metals.** Under the variation of the non-thermal parameter $\delta$, an AFM transition is continuously suppressed to zero temperature at a critical value $\delta_c$. Consequently, an AFM QCP appears at $\delta_c$ and the ground state changes from an AFM to a paramagnetic Fermi-liquid (FL) state. $T_N$ and $T_{FL}$ represent the AFM transition temperature and the onset temperature of the FL regime, respectively. $E_{loc}$ denotes the Kondo destruction energy scale that is associated with delocalization of the local moment. In a sufficiently low-temperature regime, $E_{loc}(\delta)$ separates the Kondo-screened regime with a large FS ($FS_L$, 4$f$-electrons delocalized, right side of the

$E_{loc}(\delta)$ line) and the Kondo-destruction regime with a small FS ($FS_S$, 4$f$-electrons localized, left side of the $E_{loc}(\delta)$ line). The ground state is divided into three phases: an AFM phase with a small FS ($AFM_S$), an AFM phase with a large FS ($AFM_L$), and a paramagnetic phase with a large FS ($P_L$). **a** For the conventional SDW type QCP, $E_{loc}(\delta)$ terminates inside the AFM regime. **b** For a Kondo-breakdown type QCP, $E_{loc}(\delta)$ terminates at the AFM QCP. Top and bottom insets show cartoon pictures of the FS and the spin fluctuations in the different phases. Olive, orange, and red arrows indicate the local moments, the itinerant conduction electrons, and the magnetic moments screened by the conduction electrons, respectively.

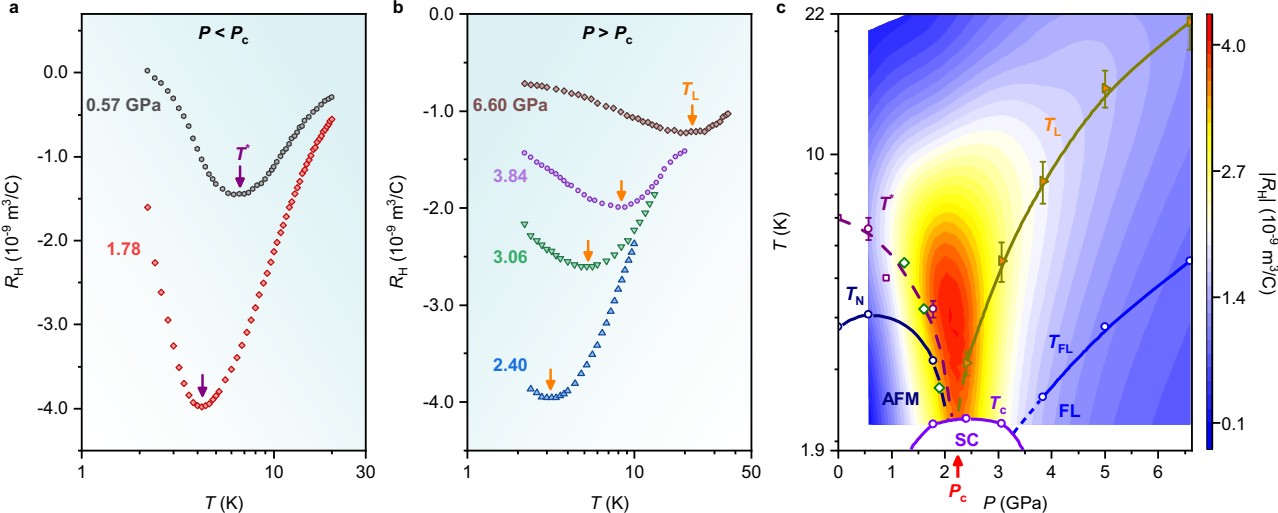

**Fig. 2 | Evolution of the Hall coefficient with pressure and the $T$–$P$ phase diagram for CeRhIn$_5$. a**, **b** Temperature dependence of Hall coefficient $R_H$ for CeRhIn$_5$ at representative pressures measured at temperatures above the superconducting transition temperature $T_c$ and under a magnetic field of 1 T applied along the $c$-axis. The field is much lower than those required to observe quantum oscillations[15] and to suppress superconductivity[21]. The purple and orange arrows represent the onset of short-range AFM spin correlations at $T^*$ and the $4f$-electron delocalization crossover temperature $T_L$, respectively (see text for details). **c** $T$–$P$ phase diagram of CeRhIn$_5$ at 0 T overlaid with a contour plot of the amplitude of the Hall coefficient $|R_H|$ at 1 T. $T^*$, $T_N$, $T_L$, $T_c$, and $T_{FL}$ are denoted by purple circles, navy circles, orange

triangles, violet circles, and blue circles, respectively. Kondo breakdown and the AFM QCP coincide at the critical pressure $P_c$ where $T_c$ reaches a maximum, the Fermi surface reconstructs from small-to-large, and temperature $T_L$ of the local extremum in $|R_H|$ extrapolates to zero temperature. The purple squares and olive diamonds are data adopted from Hall[24] and nuclear quadrupole resonance[26] measurements, respectively. The dashed and solid lines are guides to the eyes. AFM, SC, and FL stand for antiferromagnetic, superconducting, and Fermi liquid regions, respectively. Error bars on the $T^*$ and $T_L$ represent the uncertainties in determining the minimum in the Hall coefficient.

pressure-dependence of an ordinary Hall contribution at $P > P_c$ and to a change in the character of charge carriers at $T_L$.

A color-contour map of the absolute value of $R_H$ ($|R_H|$) for CeRhIn$_5$ is displayed in the temperature–pressure ($T$–$P$) plane in Fig. 2c. In the low-pressure AFM regime, $T^*$ and $T_N$ smoothly extrapolate to $T = 0$ K at $P_c$, at which $T_c$ and $|R_H|$ are a maximum. The existence of an AFM QCP at $P_c$, hidden by the pressure-induced superconducting dome, has been revealed explicitly by applying a magnetic field sufficient to suppress superconductivity[21,22]. As shown in the figure, $T_L(P)$ extrapolates smoothly on the paramagnetic side of the diagram to $T = 0$ K at the magnetic QCP where, in the limit $T \to 0$ K, the FS reconstructs, de Haas-van Alphen (dHvA) frequencies increase, and the quasiparticle effective mass diverges[15]. These are all essential characteristics of a Kondo-breakdown QCP. The phase diagram in Fig. 2c coincides with the theoretically predicted phase diagram (Fig. 1b) if we identify $T_L(P)$ as $E_{loc}(P)$, i.e., a change in the nature of charge carriers at $T_L$ is accompanied by a crossover from small-to-large Fermi surface with increasing pressure at finite temperature. Theoretically, $E_{loc}$ appears at some temperature below the onset of heavy quasiparticle formation[6], typically taken experimentally to be signaled by the temperature $T_{max}$ where the magnetic resistivity reaches a maximum. In CeRhIn$_5$, $T_{max}(P)$ is roughly 6.5 times $T_L(P)$ at $P > P_c$[28]. Multiple experiments, including dHvA measurements[15], suggest that CeRhIn$_5$ at $P = P_c$ is equivalent to the isostructural heavy-fermion superconductor CeCoIn$_5$ at $P = 0$ GPa[29]. Recent Hall measurements argue that CeCoIn$_5$ at $P = 0$ GPa is very close to a QCP characterized by a localization/delocalization of $4f$-electrons at a transition connecting two Fermi surfaces of different volumes[30]. A minimum in its Hall coefficient evolves with pressure increasing from $P = 0$ GPa in very much the same way as $T_L(P)$ shown in Fig. 2b for CeRhIn$_5$ at $P > P_c$[24]. Further, the pressure-dependent temperature of a Hall minimum tracks the delocalization crossover temperature determined by nuclear quadrupole resonance measurements in CeIn$_3$[17,31], providing additional support for associating the Hall minimum in CeRhIn$_5$ and CeCoIn$_5$ with a delocalization crossover at finite temperature. Finally, similar to $T^*$[27], the influence of magnetic field on $T_L$

(Supplementary Fig. 1) is negligible even though $|R_H|$ is suppressed, an observation arguing against a substantial contribution of magnetic fluctuations to determining $T_L$. Each of these further compels the identification of $T_L$ with $E_{loc}$.

## Quantum criticality in Sn-doped CeRhIn$_5$

We turn to the case of CeRhIn$_5$ with a Sn concentration of 0.044, CeRh(In$_{0.956}$Sn$_{0.044}$)$_5$, labelled as Sn-doped CeRhIn$_5$ in the following. Figure 3a, b shows the temperature dependence of the Hall coefficient $R_H$ for Sn-doped CeRhIn$_5$ at representative pressures up to 2.25 GPa. $R_H$ was obtained by applying a magnetic field of 5 T that completely suppresses superconductivity over the whole pressure range studied. Similar to pure CeRhIn$_5$, two characteristic temperatures $T^*$ and $T_L$ are revealed by a local minimum in $R_H$. In the low-pressure regime, $T^*$ decreases monotonically with increasing pressure and becomes unresolvable at 1.0 GPa (Fig. 3a). At pressures higher than 1.0 GPa, another local minimum in $R_H$ appears at $T_L$ and increases with increasing pressure (Fig. 3b). We note that the field effects on $T^*$ and $T_L$ in Sn-doped CeRhIn$_5$ are negligible (Supplementary Fig. 3). A color-contour map of the amplitude of $R_H$ ($|R_H|$) for Sn-doped CeRhIn$_5$ at 5 T is shown in the $T$–$P$ plane in Fig. 3c, which is overlaid with the phase boundaries. The slight Sn doping leads to a decrease of $T_N$ from 3.8 K for pure CeRhIn$_5$ to 2.1 K in the Sn-doped case at ambient pressure[32]. With applying pressure, $T_N$ decreases gradually and extrapolates to a terminal critical pressure $P_{c2}$ (~1.3 GPa) where pressure-induced superconductivity reaches a maximum $T_c$. Accompanying the suppression of $T_N$, $T^*(P)$ decreases with pressure and also extrapolates to $T = 0$ K at $P_{c2}$, a response qualitatively similar to pure CeRhIn$_5$ in which $T^*(P)$ is associated with the development of short-range AFM correlations. With $T_L(P) \ll T_{max}(P)$ (Supplementary Fig. 5 and $\partial T_L/\partial P > 0$ at $P \ge$ 1.2 GPa (Fig. 3b), as in CeRhIn$_5$ at pressures above $P_c$, we associate $T_L(P)$ with $E_{loc}(P)$ in Sn-doped CeRhIn$_5$. In contrast to CeRhIn$_5$, $E_{loc}(P)$ extrapolates to $T = 0$ K at a distinctly lower pressure $P_{c1}$ (~1.0 GPa) than the critical pressure $P_{c2}$ where AFM transition is suppressed to $T = 0$ K. The termination of $E_{loc}$ at $P_{c1}$ indicates that a local-to-itinerant

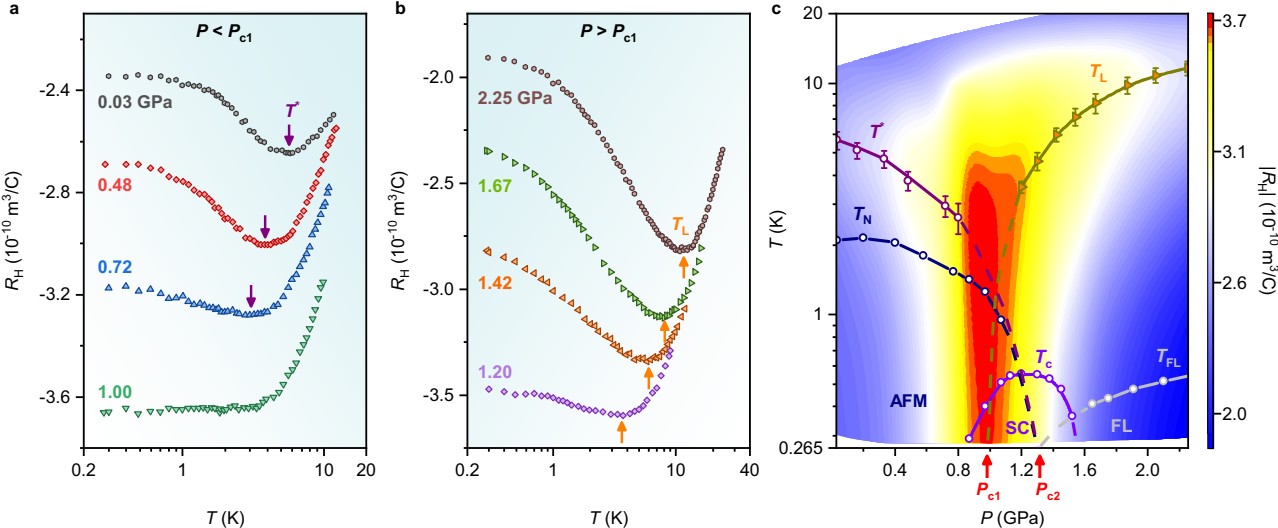

**Fig. 3 | Evolution of the Hall coefficient with pressure and the $T$–$P$ phase diagram for Sn-doped CeRhIn$_5$. a, b** Hall coefficient $R_H$ of Sn-doped CeRhIn$_5$ measured at 5 T is plotted as a function of temperature at representative pressures. The purple and orange arrows represent the onset of short-range magnetic correlations at $T^*$ and the $4f$-electron delocalization crossover temperature $T_L$, respectively (see text for details). **c** $T$–$P$ phase diagram of Sn-doped CeRhIn$_5$ at 0 T overlaid with a contour plot of the amplitude of the Hall coefficient $|R_H|$ at 5 T. $T^*$, $T_N$, $T_L$, $T_c$, and $T_{FL}$ are denoted by purple circles, navy circles, orange triangles, violet circles, and gray

circles, respectively. $P_{c2}$ (-1.3 GPa) is the critical pressure where $T_N$ extrapolates to zero temperature, corresponding to an SDW QCP, and $T_c$ reaches a maximum. $P_{c1}$ (-1.0 GPa) is another critical pressure where $T_L(P)$ extrapolates to zero temperature, indicating that destruction of the Kondo effect occurs within the AFM phase. The dashed and solid lines are guides to the eyes. AFM, SC, and FL stand for antiferromagnetic, superconducting, and Fermi liquid regions, respectively. Error bars on the $T^*$ and $T_L$ represent the uncertainties in determining the minimum in the Hall coefficient.

transformation of $4f$ degrees-of-freedom and concomitant reconstruction of FS take place within the AFM state of the Sn-doped material.

The existence of two critical pressures in Sn-doped CeRhIn$_5$ is supported by the low-temperature resistivity $\rho_{ab}$ measured parallel to the Ce-In plane under a magnetic field of 4.9 T (Supplementary Fig. 6). The color contour of isothermal resistivity in the $T$–$P$ plane, illustrated in Fig. 4a, shows a funnel of enhanced scattering centered at $P_{c1}$. The local temperature exponent $n$ derived from $n = \partial(\ln\Delta\rho)/\partial(\ln T)$, in contrast, reveals a funnel of non-Fermi-liquid behavior centered near $P_{c2}$ where the resistivity exhibits a linear-in-$T$ dependence, illustrated in Fig. 4b. Figure 4d, g shows $\rho_{ab}$ as a function of temperature at representative pressures of 0.20, 1.38, 1.45, and 2.30 GPa, respectively. A Landau-Fermi-liquid $T^2$ dependence is observed at low- and high-pressure regimes in Fig. 4d, g, but the linear-$T$ dependence is prominent near $P_{c2}$ in Fig. 4e, f. Figure 4c summarizes the dependence on pressure of the residual resistivity $\rho_0$ on the left ordinate and the temperature coefficient $A$ on the right ordinate estimated from a fit to $\rho_{ab} = \rho_0 + AT^n$. With increasing pressure, $\rho_0$ increases by over a factor of two, reaches a maximum at 1.0 GPa (=$P_{c1}$), and decreases with increasing pressure, reflecting enhanced scattering of critical charge fluctuations around $P_{c1}$[8–11]. The coefficient $A$, which is related to the effective mass of quasiparticles[19,33], however, gradually increases at low pressures, goes through a local minimum at $P_{c1}$, and peaks sharply at $P_{c2}$, which is consistent with the divergence of effective mass predicted at an SDW QCP.

## Discussion

Despite several quantum-critical heavy-fermion candidates, the delocalization energy scale $E_{loc}$ has been identified in a limited number of compounds, including YbRh$_2$Si$_2$[13,14,18], Ce$_3$Pd$_{20}$Si$_6$[34], and CeIn$_3$[17,31]. In the case of YbRh$_2$Si$_2$, $E_{loc}$ manifests as anomalies in isothermal measurements, such as a step-like crossover in the field-dependent Hall coefficient and magnetoresistivity that sharpens with decreasing temperature or a smeared kink in the field-dependent Hall resistivity, magnetostriction, and magnetization[13,14,18]. The locus of $E_{loc}$ points extrapolates to a field-tuned zero-temperature boundary of magnetic

order, as in CeRhIn$_5$ under applied pressure. Similar analysis of Hall data for Ce$_3$Pd$_{20}$Si$_6$ finds, however, that $E_{loc}$ extrapolates inside the ordered part of its field-dependent phase diagram[34], analogous to Sn-doped CeRhIn$_5$ (Fig. 3c). This also is the case with pressure-tuned CeIn$_3$, discussed earlier, where a localization/delocalization scale intersects its antiferromagnetic phase at finite temperature[17,31].

In these other examples, the signature for $E_{loc}$ in various physical quantities has been used to infer a Fermi-surface change, but dHvA measurements unambiguously establish a jump in dHvA frequencies at the critical pressure where $T_L$ extrapolates to $T = 0$ K in CeRhIn$_5$. A simple interpretation of $R_H$ would anticipate an associated step-like jump in carrier density at $P_c$, but as shown in Fig. 5a, instead of having a sharp jump, the pressure-dependent isothermal Hall coefficient peaks strongly at $P_c$. Perhaps measurements have not been made at sufficiently low temperatures to reveal a jump. More likely it is obscured by effects of critical charge and spin fluctuations on the Hall resistivity and/or only a small net change in the sum of hole and electron contributions in this nearly compensated metal. Though present experiments cannot make a definitive distinction, they do reflect the approach to FS reconstruction detected in dHvA that is coincident with a magnetic QCP. In contrast to the sharp peak in $|R_H(P)|$ in CeRhIn$_5$, there is a broad maximum $|R_H(P)|$ that peaks at $P_{c1}$ but encompasses both $P_{c1}$ and $P_{c2}$ in CeRh(In$_{0.956}$Sn$_{0.044}$)$_5$ (Fig. 5b). Such a broad maximum relative to that in CeRhIn$_5$ is not surprising because of the close proximity of two critical pressures, each with their own spectrum of critical spin/charge fluctuations, and smearing these spectra by disorder inherent to the Sn substitution.

The distinctly different critical behaviors of pure and Sn-doped CeRhIn$_5$ are captured in a generalized quantum-critical phase diagram that includes both Kondo-breakdown and SDW criticality[8,35]. In this theory, the nature of quantum criticality is determined by two quantities, $\delta$ and $G$ (Supplementary Fig. 9), where, as before, $\delta = k_B T_K/I$, and $G$ is a parameter that reflects magnetic frustration or effective spatial dimensionality. Our results are consistent with Kondo breakdown coinciding in pure CeRhIn$_5$ under pressure with a $T = 0$ K transition from an AFM state with a small FS (AFM$_S$) to a paramagnetic state with

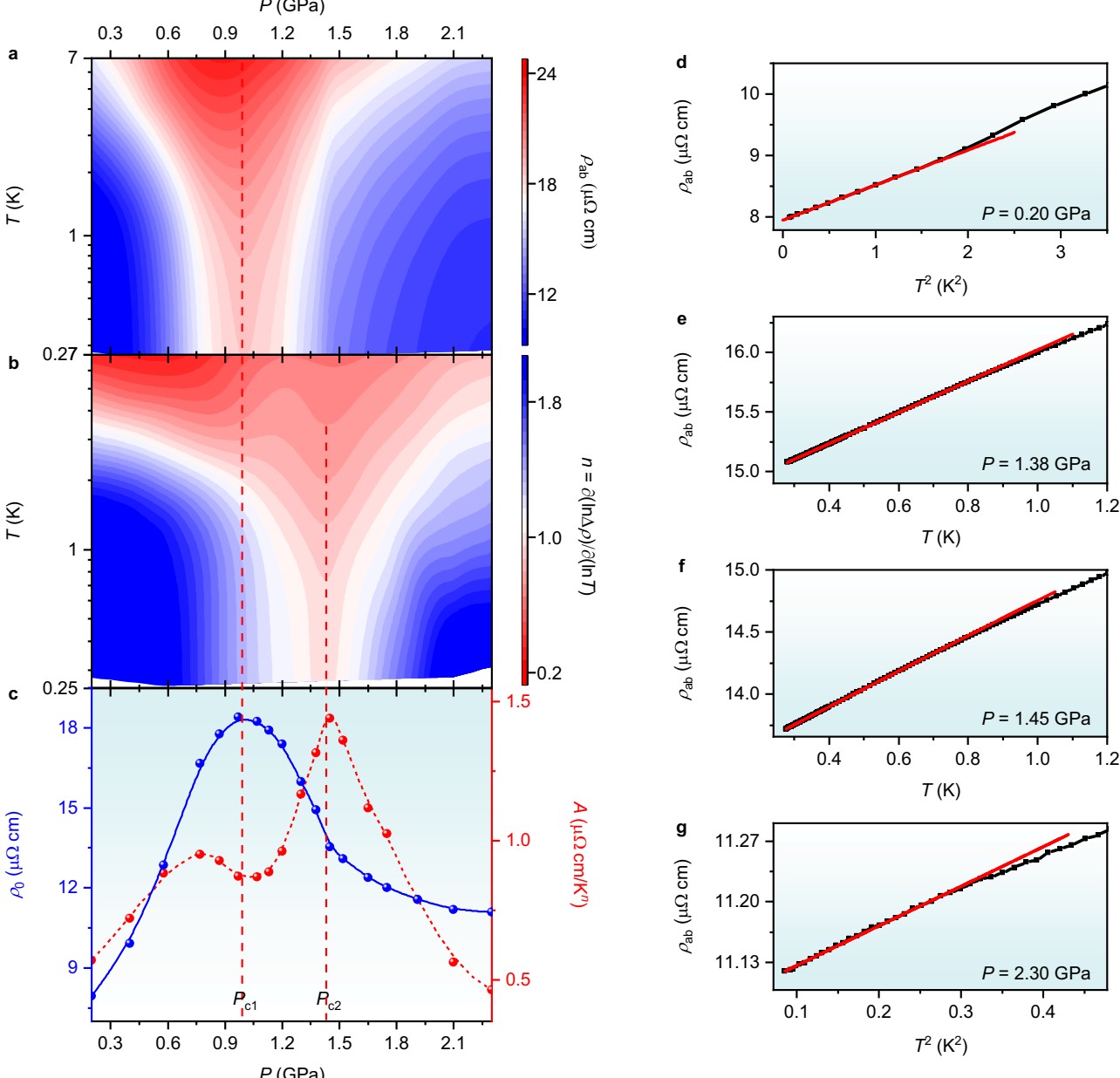

**Fig. 4 | Quantum criticality of Sn-doped CeRhIn$_5$ under pressure. a** Contour plot of resistivity $\rho_{ab}$ for Sn-doped CeRhIn$_5$ measured parallel to the Ce-In plane under a magnetic field of 4.9 T. The significant enhancement of $\rho_{ab}$ is centered around $P_{c1}$. **b** Colours represent the local temperature exponent $n$ derived from $n = \partial(\ln\Delta\rho)/\partial(\ln T)$, where $\Delta\rho = \rho_{ab} - \rho_0 = AT^n$ and $\rho_0$ is the residual resistivity. A funnel regime with linear-$T$ dependence of $\rho_{ab}$ is observed around $P_{c2}$, a characteristic of the non-Fermi liquid behavior near the AFM QCP, as shown in **e**, **f** at representative

pressures of 1.38 and 1.45 GPa, respectively. **c** Pressure dependence of the residual resistivity $\rho_0$ (left-axis, blue circles) and the coefficient $A$ (right-axis, red circles) determined by fitting the low-temperature resistivity to $\rho_{ab} = \rho_0 + AT^n$. **d**–**g** show fits to representative data from which **c** is constructed. The dashed red lines in **a**–**c** are guides to the eyes. The red lines in **d**–**g** are least-squares fits to the low-temperature data.

a large FS ($P_L$) at $P_c$, i.e., Kondo breakdown and the AFM QCP coincide. Similar to Sn-doped CeCoIn$_5$[36], Sn substitution for In in CeRhIn$_5$ enhances hybridization between $4f$- and conduction electron wave functions, i.e., increases $T_K$ and thus $\delta$. Stronger hybridization also reduces frustration among magnetic exchange pathways[8,37]. Analysis of magnetic neutron-diffraction experiments finds a clear change in magnetic structure and decrease of the ordered moment for a Sn concentration $x \geq 0.052$, comparable to CeRh(In$_{0.956}$Sn$_{0.044}$)$_5$ under a modest pressure, that is attributed to an abrupt modification of the Fermi surface[38]. With these changes induced by Sn doping, the system follows a different trajectory that goes through an intermediate magnetically ordered state with a large FS (AFM$_L$), i.e., Kondo breakdown

occurs inside the AFM region, and thus the corresponding AFM QCP is of the SDW type as illustrated in Fig. 1a and Supplementary Fig. 9 and found in the prototypical Kondo-breakdown system YbRh$_2$Si$_2$ when Rh is replaced by a small amount of Co[18].

Before identifying the crossover scale $E_{loc}$ in pure and Sn-doped CeRhIn$_5$, their criticality was ambiguous, possibly either Kondo-breakdown or SDW[32,39]. That ambiguity now is removed, with consequences for an interpretation of the origin of their pressure-induced superconductivity. At the SDW QCP in Sn-doped CeRhIn$_5$, which is decoupled from the Kondo breakdown and where $T_c$ reaches its maximum, quantum fluctuations of the magnetic order parameter are the prime candidate for mediating Cooper pairing[40]. Fluctuations

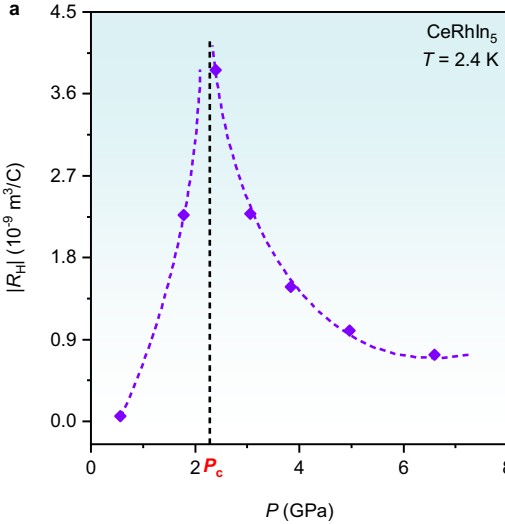

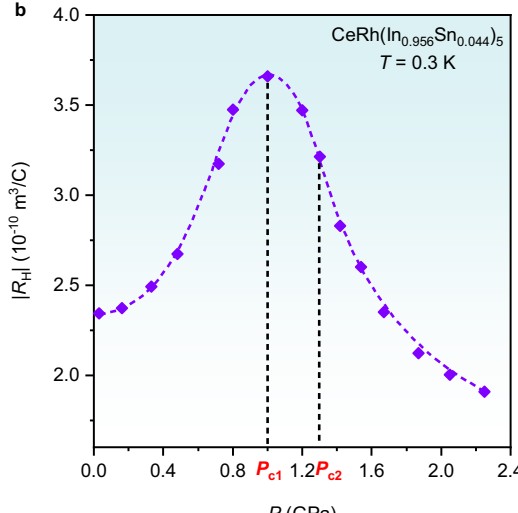

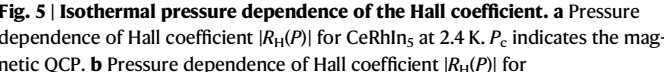

**Fig. 5 | Isothermal pressure dependence of the Hall coefficient. a** Pressure dependence of Hall coefficient $|R_H(P)|$ for CeRhIn$_5$ at 2.4 K. $P_c$ indicates the magnetic QCP. **b** Pressure dependence of Hall coefficient $|R_H(P)|$ for CeRh(In$_{0.956}$Sn$_{0.044}$)$_5$ at 0.3 K. $P_{c2}$ denotes the magnetic QCP. $P_{c1}$ indicates the critical pressure where $T_L$ extrapolates to zero temperature and is lower than the magnetic QCP $P_{c2}$.

around a Kondo-breakdown QCP, however, are far more complex and involve not only quantum-critical fluctuations of a magnetic order parameter but also of the Fermi surface, i.e., charge degrees-of-freedom[8–11]. Initial model calculations show that critical Kondo-breakdown fluctuations can produce a superconducting instability[41,42], but much remains to make these calculations directly testable by experiment. Interestingly, $T_c \approx 2.3$ K of CeRhIn$_5$ at $P_c$ and of CeCoIn$_5$ at $P = 0$ GPa is among the highest of any rare-earth-based heavy-fermion superconductor.

The theory of Kondo-breakdown criticality in heavy-fermion materials has two essential signatures—an abrupt change from small-to-large Fermi surface coincident with magnetic criticality and the charge delocalization crossover scale $E_{loc}$ that extrapolates from the paramagnetic state to the QCP[6]. Without both, Kondo-breakdown criticality cannot be established with certainty. Our Hall measurements provide compelling evidence for $E_{loc}$ in pure and Sn-doped CeRhIn$_5$ and, in light of dHvA results[15], for Kondo-breakdown criticality in CeRhIn$_5$. Finally, we speculate that critical charge fluctuations at a Kondo-breakdown QCP, evidenced by $\omega/T$ scaling of the frequency ($\omega$)-dependent optical conductivity[10], should play a non-trivial role in these signatures for $E_{loc}$. Making this connection experimentally and theoretically would mark a significant advance.

## Methods
Single crystals of pure and Sn-doped CeRhIn$_5$ were synthesized using the standard self-flux technique[32]. The high-pressure resistivity and Hall measurements on CeRhIn$_5$ were carried out using the Van der Pauw method[43] in a diamond-anvil cell made of Be-Cu alloy. NaCl powder was applied as the pressure medium to obtain a quasihydro-static pressure environment. The pressure in the diamond-anvil cell was determined by the ruby fluorescence method[44]. The high-pressure resistivity and Hall measurements on CeRh(In$_{0.956}$Sn$_{0.044}$)$_5$ were measured using the standard six-probe method in a Be-Cu/NiCrAl hybrid clamp-type cell. Daphne oil was employed as the pressure medium to obtain a hydrostatic pressure environment. The pressure dependence of the superconducting transition temperature of Pb was used to determine the pressure inside the clamp-type cell[45]. All measurements were performed with a low-frequency resistance bridge from Lake Shore Cryotronics. A $^4$He cryostat without magnetic field and a Physical Property Measurement System with a maximum magnetic field of 9 T were used in the temperature range of 1.8 to 300 K. A

HelioxVL system with a maximum magnetic field of 12 T was used to control temperature down to 0.3 K.

## Data availability
All data supporting the findings of this study are available within the article and the supplementary information. The data are available upon request to the corresponding authors.

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

## Acknowledgements

This work was supported by Basic Science Research Program through the National Research Foundation of Korea (NRF) funded by the Ministry of Education (No. 2021R1I1A01047499 (H.W.)) and by the National Research Foundation of Korea (NRF) grant funded by the Korea government (MSIT) (No. 2021R1A2C2010925 (T.P.) and No. RS-2023-00220471 (T.P.)). Work at Los Alamos was performed under the auspices of the U.S. Department of Energy, Office of Basic Energy Sciences, Division of Materials Science and Engineering.

## Author contributions

All authors discussed the results and commented on the manuscript. T.P. and H.W. conceived the study. T.B.P. and H.W. performed the measurements and analyzed the data. J.K. and H.J. assisted the transport measurements setup. T.B.P. and E.D.B. synthesized pure and Sn-doped CeRhIn$_5$ single crystals. H.W., T.B.P., T.P., and J.D.T. wrote the manu-script with input from all authors.

## Competing interests

The authors declare no competing interests.
