## [Peer Review File · Nature Communications]

REVIEWER COMMENTS

Reviewer #1 (Remarks to the Author):

In the manuscript “Charge delocalization crossover in the quantum critical superconductor CeRhIn5” by H. Wang et al, the authors have performed resistivity measurements under applied pressure of both the stoichiometric CeRhIn5 and its Sn-doped analog at a particular doping $x=0.044$. By following certain features in transport data – the positions in temperature T_L of the local maxima of the Hall coefficient $|RH(T)|$, the authors claim to have identified the energy scale E_{loc} that signals the small-to-large Fermi surface crossover (or equivalently, the local-to-itinerant magnetism crossover). Based on these data, the authors study the pressure dependence of $E_{loc}(P)$, finding that it extrapolates to zero temperature at a pressure P_{c1} which is qualitatively different between the stoichiometric CeRhIn5 and its Sn-doped analog. In the former case, the authors claim P_{c1} coincides with the magnetic QCP (where $T_{Neel} \rightarrow 0$), whereas in the latter, the claim is that P_{c1} lies to the left of the QCP, inside the magnetic dome. The interpretation that the authors provide, is that the former behaviour in CeRhIn5 is consistent with the scenario of the so-called Kondo breakdown, whereas the latter results in Sn-doped compound indicate a more conventional “spin-density wave” quantum criticality.

The manuscript is carefully written and results presented clearly. However, I cannot recommend it for publication in the present form because of two crucially important issues:

(i) it is impossible to verify the author’s claim that $P_{c1} \neq P_{c2}$ from the available data in Sn-doped compound because this hinges on a qualitative (“guide to the eye”) extrapolation of the TL and TN curves to zero temperature, fraught with uncertainty, and (ii) the identification of the maximum of Hall coefficient as a function of temperature with the position of the local-to-itinerant crossover scale E_{loc} is tenuous at best, for which the authors have provided no physical justification. I elaborate on these points below.

(i) Even if one were to accept the authors’ claim about the identification of T_L with E_{loc} at face value (more on that below), the matter of fact is that the left-most datapoint for T_L in Fig. 3c appears at the pressure 1.2 GPa and $T_L = 4K$, and the authors extrapolate this rather frivolously to $T_L = 0$ at the pressure $P_{c2} = 1.0$ GPa. The situation is equally dubious with the Neel temperature, for which the last datapoint in Fig. 3c is given by $T_N=1K$ at $P=1.1$ GPa. With no datapoints reported at lower temperatures, it is conceivable that the Neel transition becomes weakly first order and terminates at the same pressure, around 1.15 GPa, where the TL line abruptly dives to zero. This would be completely consistent with the first-order jump from the small- to large Fermi surface volume detected in the stoichiometric CeRhIn5 by Shishido et al. And it follows therefore that there is no true evidence, other than “guide to the eye” extrapolation in Fig. 3c, that Sn-doped compound

is any different from the stoichiometric one, where P_{c1} and P_{c2} are claimed to coincide. This then puts to question the entire set of conclusions that the authors claim to have found.

I can be persuaded if the authors could show the data at 1.15 GPa, showing that either T_N , or T_L (or both) remain finite. Can the authors please perform this key measurement?

(ii) The most puzzling part of the authors' analysis is that of the Hall data in Figs 2b and 3b. Why should one associate a maximum of $|RH(T)|$ with the local-to-itinerant crossover scale E_{loc} ? Yes, I've read the author's explanation on pp.4-5, but it's suggestive at best, and potentially quite misleading. Indeed, the value of $|RH|$ is non-monotonic in pressure, as Fig. 3b clearly shows, and the $T \rightarrow 0$ extrapolated value of $|RH|$ at $P=0.48$ GPa in the localized regime is actually lower, in absolute value, than at, say, $P=1.2$ GPa to the right of P_{c2} , in what should be the large-Fermi surface regime. I realize that interpreting the Hall coefficient in multi-band systems is tricky, but assuming that the carrier mobilities are temperature independent at these low temperatures below ~ 10 K, the consistent logical explanation is that the carrier density $n_{loc}(P < P_{c2})$ is higher than $n_{itin}(P > P_{c2})$. In other words, that would mean that there are *more* carriers in the localized, small FS regime than in the itinerant phase at $P > P_{c2}$! How could that be?

I would like to contrast this with the situation in $YbRh_2Si_2$ where the Kondo breakdown scenario was established using the Hall measurements. There, the Hall coefficient *decreases* monotonically as a function of control parameter (in that case, magn. field) upon tuning from the localized to itinerant regime, see Fig. 2b in Ref. 13 by Paschen et al. That is fully consistent with $n_{loc} < n_{itin}$.

I strongly suggest that the authors address this issue head-on, but plotting the absolute value of the Hall coefficient $|RH|$ along the isotherms as a function of pressure – essentially the analog of Fig. 2b in Ref. 13. I realize that the authors only have a few discrete values of pressure at their disposal, but still, their Fig.3b contains at least 14 pressure datapoints – that should be more than enough to plot the graph of $|RH(P)|$ along the isotherms. Would the resulting graph be monotonic? Would the authors observe the sharpening of the transition as the temperature lowered, the way it is seen in $YbRh_2Si_2$? These are crucially important questions that beg to be answered, but it's very difficult to do so based solely on the color plots in Fig. 2c and 3c, it would have been much more informative if the authors provided the data on the pressure dependence as I request here.

(iii) Perhaps a more minor point: on page 6, when analyzing the temperature dependence of the resistivity, the authors associate the A-coefficient of the power-law dependence ($\rho = \rho_0 + AT^n$) with the effective mass of the quasiparticles. This is only true in the Fermi liquid regime $n=2$ where $A \sim (m^*)^2$, however it is not clear what basis there is, if any, to adopt the same treatment for the non-FL regime near P_{c2} , where the quasiparticles might not even be well defined! In fact, one could have fitted the resistivity in that regime as $\rho(T) = \rho_0 + AT^2 + BT^1$, and the data would suggest that the Fermi-liquid coefficient A would vanish at P_{c2} , rather than diverge, yielding the opposite conclusion regarding the effective mass. Can the authors explain why such analysis cannot be adopted?

Reviewer #2 (Remarks to the Author):

The manuscript deals with charge carrier delocalization transitions across (magnetic) quantum critical points, that appear to be relevant to many classes of strongly correlated electron systems and to be essential for understanding high-temperature superconductivity. The material under investigation is the well-studied heavy fermion compound CeRhIn₅. A previous de Haas-van Alphen (dHvA) study (Ref. 15 of the manuscript) indicated that in this material a drastic change of the Fermi surface occurs at a critical pressure of 2.3 GPa. A jump in the dHvA frequency and a substantial mass enhancement towards the critical pressure were extracted using magnetic fields between 10 and 16.9 T.

The present work uses Hall effect measurements to scrutinize these results. The measurements are done in a much smaller magnetic field (1 T) which has the benefit that the material is much closer to the B=0 quantum critical point. The claim is that the Hall effect measurements allow identifying “an energy scale E_{loc} that signals a local-to-itinerant crossover of 4f degrees-of-freedom and terminates at the antiferromagnetic QCP”, thereby “establishing the origin of the Fermi surface reconstruction observed in the quantum oscillation measurements and the Kondo-breakdown nature of quantum criticality”.

I am not convinced that the presented data allow drawing such strong conclusions.

The experimental signatures that the authors use are minima in the temperature-dependent Hall coefficient $R_H(T)$, measured at different fixed pressures across the temperature-pressure phase diagram of CeRhIn₅ and of Sn-doped CeRhIn₅. $T^*(p)$ is the curve connecting the temperatures of the R_H minima below a critical pressure, $T_L(p)$ above it.

As sketched in Fig. 1, the Fermi surface is expected to differ on the two sides of E_{loc} . Ideally, this is probed in the $T=0$ limit as a function of the tuning parameter δ , as the notion of the Fermi surface is sharply defined only at $T=0$. In such an experiment, the Hall coefficient should exhibit a jump at δ_c in the zero temperature limit. It is not clear to me why this would translate into minima in $R_H(T)$ curves. It is clear that with decreasing temperature the “large” Fermi surface builds up very gradually, which may or may not be reflected in a change of R_H (there are many potential sources of temperature-dependent Hall effects in multiband heavy fermion systems). But on the other side of E_{loc} , where the “small” Fermi surface exists, such a change would then not be expected, but still is observed in the experiments.

To me, it seems that the minima in $R_H(T)$ (and thus the $T^*(p)$ and $T_L(p)$ curves) are indicative of the boundaries of the quantum critical fan but cannot necessarily inform us whether or not the Hall effect will jump (as a function of p) in the zero temperature limit.

I would like to encourage the authors to plot (from their $R_H(T)$ data at fixed p) $R_H(p)$ curves at a set of different fixed temperatures. Is a Hall crossover seen? If yes, at which pressure? And does it sharpen with decreasing temperature to be indicative of a jump in the $T=0$ limit?

Another point is whether the authors have firm evidence that the AFM phase transition remains continuous up to p_c at the field of $B=1T$ (used in their R_H measurements).

I'd be happy to be convinced that the author's (very interesting!) conclusions are robust, but do judge the evidence put forward in the present manuscript insufficient. So, unfortunately, I cannot recommend publication in the present form.

Reviewer #3 (Remarks to the Author):

Wang et al. present Hall effect and resistance measurements on $CeRhIn_5$ and Sn-doped $CeRhIn_5$ and claim that these demonstrate the separation of a local energy scale from the magnetic quantum critical point. The results are of interest to a large community of researchers working on quantum criticality and underpins research in the wider area of correlated materials. I recommend revision of this manuscript before it is accepted for publication.

Whilst the experimental data are of excellent quality, the interpretation of the results should be clarified. The authors associate extrema in the temperature dependence of the Hall coefficient with the magnetic fluctuations at low pressures and with a Fermi surface reconstruction at high pressures. Arguments are provided that the high-pressure extrema are not associated with magnetic fluctuations based on the insensitivity to magnetic field. However, it would be good to also discuss and show evidence for the low-pressure extrema to be associated with spin fluctuations, i.e. T^* . Is there a larger sensitivity to magnetic field for the low-pressure extrema? In fact, it appears that ref 24 has observed a large field dependence of the Hall coefficient at 2 GPa. Has this been reproduced by the present study?

Can the authors comment on the isothermal pressure dependence of the Hall coefficient. It appears from their contour plots that this will display a single maximum. How can this be reconciled with two separate energy scales?

We would like to thank referees for the suggestions and comments on our manuscript. In order to address the concerns raised by the referees, we have performed additional measurements and provided more comprehensive analysis. Below are our point by point replies to the comments.

Reply to Referee #1

In the manuscript “Charge delocalization crossover in the quantum critical superconductor CeRhIn₅” by H. Wang et al, the authors have performed resistivity measurements under applied pressure of both the stoichiometric CeRhIn₅ and its Sn-doped analog at a particular doping $x = 0.044$. By following certain features in transport data – the positions in temperature T_L of the local maxima of the Hall coefficient $|R_H(T)|$, the authors claim to have identified the energy scale E_{loc} that signals the small-to-large Fermi surface crossover (or equivalently, the local-to-itinerant magnetism crossover). Based on these data, the authors study the pressure dependence of $E_{loc}(P)$, finding that it extrapolates to zero temperature at a pressure P_{c1} which is qualitatively different between the stoichiometric CeRhIn₅ and its Sn-doped analog. In the former case, the authors claim P_{c1} coincides with the magnetic QCP (where $T_{Nee1} \rightarrow 0$), whereas in the latter, the claim is that P_{c1} lies to the left of the QCP, inside the magnetic dome. The interpretation that the authors provide, is that the former behaviour in CeRhIn₅ is consistent with the scenario of the so-called Kondo breakdown, whereas the latter results in Sn-doped compound indicate a more conventional “spin-density wave” quantum criticality.

The manuscript is carefully written and results presented clearly. However, I cannot recommend it for publication in the present form because of two crucially important issues: (i) it is impossible to verify the author’s claim that $P_{c1} \neq P_{c2}$ from the available data in Sn-doped compound because this hinges on a qualitative (“guide to the eye”) extrapolation of the T_L and T_N curves to zero temperature, fraught with uncertainty, and (ii) the identification of the maximum of Hall coefficient as a function of temperature with the position of the local-to-itinerant crossover scale E_{loc} is tenuous at best, for which the authors have provided no physical justification. I elaborate on these points below.

Q1: Even if one were to accept the authors' claim about the identification of T_L with E_{loc} at face value (more on that below), the matter of fact is that the left-most datapoint for T_L in Fig. 3c appears at the pressure 1.2 GPa and $T_L = 4$ K, and the authors extrapolate this rather frivolously to $T_L = 0$ at the pressure $P_{c2} = 1.0$ GPa. The situation is equally dubious with the Neel temperature, for which the last datapoint in Fig. 3c is given by $T_N = 1$ K at $P = 1.1$ GPa. With no datapoints reported at lower temperatures, it is conceivable that the Neel transition becomes weakly first order and terminates at the same pressure, around 1.15 GPa, where the T_L line abruptly dives to zero. This would be completely consistent with the first-order jump from the small- to large Fermi surface volume detected in the stoichiometric CeRhIn₅ by Shishido et al. And it follows therefore that there is no true evidence, other than "guide to the eye" extrapolation in Fig. 3c, that Sn-doped compound is any different from the stoichiometric one, where P_{c1} and P_{c2} are claimed to coincide. This then puts to question the entire set of conclusions that the authors claim to have found. I can be persuaded if the authors could show the data at 1.15 GPa, showing that either T_N , or T_L (or both) remain finite. Can the authors please perform this key measurement?

Response: As shown in Fig.3(c), with the suppression of AFM transition, the pressure-induced superconductivity appears and gradually increases. As the AFM transition gets closer to the superconducting transition, it becomes more difficult to determine the AFM transition at 0 T above 1.07 GPa.

In order to identify T_N at higher pressure, external magnetic field of 4.9 T is applied. As shown in Fig. R1-1 below, T_{NS} at 0 T (square symbols) and 4.9 T (cross symbols) are overlapped, showing that T_N is almost independent of the magnetic field. At 1.2 GPa, which is higher than the critical pressure P_{c1} of 1.0 GPa, T_N is 0.73 K and T_L is 3.56 K. Observation of both T_N and T_L at 1.2 GPa shows that both characteristic temperatures remain finite in the critical regime between P_{c1} and P_{c2} , supporting that they are decoupled and terminate at two different pressures. Being consistent with the decoupled critical points, the maximum in the Hall coefficient and the effective mass occurs at P_{c1} and P_{c2} , respectively. In order to clarify these points, we have added Fig. R1-1 in the supplementary materials.

Fig. R1-1 (a–d) The first derivative of resistivity at 0 T (open symbols) and 4.9 T (solid symbols) for representative pressures. Inserts of (c) and (d) show an enlarged view of the first derivative of resistivity at 4.9 T in the low-temperature regime. The violet and navy arrows indicate the superconducting and AFM transitions, respectively. (e) T - P phase diagram of Sn-doped CeRhIn_5 . The navy squares and crosses represent the AFM transition determined from the first derivative of resistivity at 0 T and 4.9 T, respectively. The violet circles denote the superconducting transition temperature T_c determined by the zero-resistivity temperature. The purple circles and orange triangles represent the onset of short-range magnetic correlations at T^* and the $4f$ -electron delocalization crossover temperature T_L , obtained by the minimum of Hall coefficient $R_H(T)$ at 5 T.

Q2: The most puzzling part of the authors' analysis is that of the Hall data in Figs 2b and 3b. Why should one associate a maximum of $|R_H(T)|$ with the local-to-itinerant crossover scale E_{loc} ? Yes, I've read the author's explanation on pp.4-5, but it's suggestive at best, and potentially quite misleading. Indeed, the value of $|R_H|$ is non-monotonic in pressure, as Fig. 3b clearly shows, and the $T \rightarrow 0$ extrapolated value of $|R_H|$ at $P = 0.48$ GPa in the localized regime is actually lower, in

absolute value, than at, say, $P = 1.2$ GPa to the right of P_{c2} , in what should be the large-Fermi surface regime. I realize that interpreting the Hall coefficient in multi-band systems is tricky, but assuming that the carrier mobilities are temperature independent at these low temperatures below ~ 10 K, the consistent logical explanation is that the carrier density n_{loc} ($P > P_{c1}$). In other words, that would mean that there are more carriers in the localized, small FS regime than in the itinerant phase at $P > P_{c2}$! How could that be? I would like to contrast this with the situation in YbRh_2Si_2 where the Kondo breakdown scenario was established using the Hall measurements. There, the Hall coefficient decreases monotonically as a function of control parameter (in that case, magn. field) upon tuning from the localized to itinerant regime, see Fig. 2b in Ref. 13 by Paschen et al. That is fully consistent with $n_{loc} < n_{itin}$. I strongly suggest that the authors address this issue head-on, but plotting the absolute value of the Hall coefficient $|R_H|$ along the isotherms as a function of pressure – essentially the analog of Fig. 2b in Ref. 13. I realize that the authors only have a few discrete values of pressure at their disposal, but still, their Fig.3b contains at least 14 pressure datapoints – that should be more than enough to plot the graph of $|R_H(P)|$ along the isotherms. Would the resulting graph be monotonic? Would the authors observe the sharpening of the transition as the temperature lowered, the way it is seen in YbRh_2Si_2 ? These are crucially important questions that beg to be answered, but it's very difficult to do so based solely on the color plots in Fig. 2c and 3c, it would have been much more informative if the authors provided the data on the pressure dependence as I request here.

Response: As the referee pointed out, the Hall coefficient of YbRh_2Si_2 decreases monotonically as a function of magnetic field upon tuning from the localized to itinerant regime, indicating that $n_{loc} < n_{itin}$ in this case. In CeRhIn_5 , where the sudden change of Fermi surface from small to large has been detected at a critical pressure of 2.3 GPa (Ref. 15), however, a maximum of Hall coefficient $|R_H|$ is observed near the QCP, showing a non-monotonic pressure dependence of $|R_H|$ (see Fig. R1-2). We can see that the value of $|R_H|$ is lower in the localized regime than that in the itinerant phase. A similar pressure dependence of $|R_H|$ is observed in Sn-doped CeRhIn_5 , as shown in Fig. R1-3, but a difference is that the maximum of $|R_H|$ appears at a lower pressure than the magnetic QCP.

Fig. R1-2 (a) Pressure dependence of Hall coefficient in CeRhIn₅. The black circles represent the maximum of $|R_H|$ obtained at 0 T (adopted from Ref. 24). Notably, although the absolute value of $|R_H|$ is affected strongly by field, the pressure dependence of $|R_H|$ will not be changed if the $|R_H|$ is chosen at same temperature. (b) Pressure dependence of Hall coefficient for CeRhIn₅ at 2.4 K.

Fig. R1-3 (a) Pressure dependence of Hall coefficient $|R_H|$ for CeRh(In_{0.956}Sn_{0.044})₅ at 0.3 K (solid red balls, right y-axis) along with the phase diagram (left y-axis). The open circles in orange and violet represent the AFM (T_N) and superconducting (T_c) transition temperatures, respectively. P_{c1} is the critical pressure where $|R_H|$ reaches a maximum. P_{c2} is the AFM QCP where T_N extrapolates to 0 K and T_c is maximum. Dashed red lines are guides to the eyes. (b) Pressure dependence of Hall coefficient $|R_H|$ for CeRh(In_{0.956}Sn_{0.044})₅ at several representative temperatures.

The deviation of the pressure dependence of $|R_H(P)|$ in pure and Sn-doped CeRhIn₅ from the ideal case may arise because the Hall effects in multiband heavy fermion systems could be affected by

different sources. In a system like CeRhIn₅, it can be influenced not only by changes in carrier density but also by scattering rates on different parts of the Fermi surface, details of the Fermi surface topology, and the presence of spin fluctuations.

We note that the appearance of AFM transition can also affect the isothermal Hall effects. For example, in Ce₃Pd₂₀Si₆, there are two crossover behavior below the AFM transition: as shown in Fig. R1-4(a), one is related with the charge delocalization as shown in YbRh₂Si₂, while the other is related with the AFM transition, which is not observed in YbRh₂Si₂. The E_{loc} can also be determined from the magnetoresistivity that exhibits a step-like crossover similar to the Hall coefficient. As shown in Fig. R1-4(b), the magnetoresistivity of Ce₃Pd₂₀Si₆ exhibits a step-like crossover above the AFM transition, which is similar to that in YbRh₂Si₂. However, below the AFM transition, the magnetoresistivity contains a background contribution, a crossover due to the delocalization, and an increase of resistivity at low magnetic fields due to the AFM transition. In this kind of compounds, other contributions need to be removed to get a crossover due to the delocalization, as shown in R1-4(c).

Fig. R1-4 (a) Isothermal Hall resistivity of Ce₃Pd₂₀Si₆ as a function of magnetic field at different temperatures below the AFM transition. (b) Isothermal magnetoresistivity of Ce₃Pd₂₀Si₆ as a function of magnetic field. (c) Representative fitting procedure for magnetoresistivity corrected for the background contribution ($\rho_l - \rho_{l,back}$). B_N and B^* represent two crossover fields. The blue dashed and red solid lines represent the contribution from the AFM transition and delocalization, respectively. Adopted from Custers et al., Nat. Mater 11, 189 (2012).

In addition to the isothermal thermodynamic and transport properties, the energy scale of E_{loc} can also be reflected in a change of the temperature dependent measurements due to the gradual development of large Fermi surface with decreasing temperature.

(a) In pure and doped YbRh_2Si_2 , the E_{loc} can be shown as: (i) an inflection in the isothermal field-dependent Hall coefficient and magnetoresistivity (Ref. 13, 18); (ii) a crossover or kink in the isothermal field-dependent magnetostriction, Hall resistivity, and magnetization (Ref. 14); (iii) a maximum in the temperature dependence of susceptibility, as shown in Fig. R1-5.

Fig. R1-5 Temperature-dependent susceptibility of $\text{Yb}(\text{Rh}_{0.94}\text{Ir}_{0.06})_2\text{Si}_2$ (a) and $\text{Yb}(\text{Rh}_{0.93}\text{Co}_{0.07})_2\text{Si}_2$ (b) at selected magnetic fields. The red arrows indicate the temperatures of the maxima assigned to the delocalization energy scale. Adopted from Friedemann et al., Nat. Phys 5, 465 (2009).

(b) In CeIn_3 , E_{loc} can be defined by: (i) a characteristic temperature at which the nuclear quadrupole resonance nuclear-spin-lattice relaxation rate $1/T_1$ deviates from its constant value (Fig. R1-6(a)); (ii) a sudden change in the nuclear quadrupole resonance frequency $\nu_Q(P)$ (Fig. R1-6(b)); (iii) a minimum in the temperature dependence of Hall coefficient $R_H(T)$ (Fig. R1-7(a)). The delocalization crossover temperature E_{loc} or T_L chosen by these measurements are consistent, as indicated by the purple and violet triangles in Fig. R1-7(b). The termination of E_{loc} within the magnetic dome is consistent with the expectation for the SDW quantum criticality.

Fig. R1-6 (a) Temperature dependence of the nuclear-spin-lattice relaxation rate $1/T_1$ for CeIn_3 at representative pressures. The dotted line indicates a relation of $1/T_1 = \text{const.}$ Dotted and solid arrows point to the AFM transition T_N and the delocalization crossover temperature T_L , respectively. (b) Pressure dependence of the nuclear quadrupole resonance frequency ν_Q at 10 K. ν_Q begins to increase significantly above 2 GPa due to the delocalization. Adopted from Kawasaki et al., *Phy. Rev. B* 77, 064508 (2008).

Fig. R1-7 (a) Temperature dependence of the Hall coefficient for CeIn_3 under high pressure. (b) T - P phase diagram of CeIn_3 . The red and blue squares indicate the AFM transition T_N and superconducting transition T_c , respectively. The violet and purple triangles indicate the delocalization crossover T_L , determined by the Hall minimum and the deviation temperature in the nuclear quadrupole resonance measurements, respectively. Adopted from Araki et al., *J. Phys. Soc. Jpn.*, 84, 123702 (2015).

These examples underline that the charge delocalization E_{loc} can be shown by different signatures depending on the systems and measurements. Considering the similarities between $CeIn_3$ and $CeRhIn_5$, the minimum in the temperature dependence of Hall coefficient $R_H(T)$ in $CeRhIn_5$ can also be a signature for E_{loc} . Especially, the Hall minimum temperature in $CeRhIn_5$ extrapolated from the paramagnetic side terminates at the magnetic QCP where a sudden change of Fermi surface from small to large has been observed by the de Haas-van Alphen measurements, which further supports the correspondence between the Hall minimum temperature and the charge delocalization crossover energy scale of E_{loc} above P_c . The nuclear quadrupole resonance measurements can be used to further confirm this the energy scale E_{loc} in pure and Sn-doped $CeRhIn_5$.

Per recommendation, we have added Fig. R1-2(b) and Fig. R1-3(b) in the supplementary materials (Fig. S9) to show the isothermal pressure dependent Hall coefficient in pure and Sn-doped $CeRhIn_5$. We have also added the discussion of E_{loc} in $CeIn_3$ in the second paragraph of section “Observation of E_{loc} in $CeRhIn_5$ ” on page 5 as another supporting evidence for the relationship between the Hall minimum temperature and E_{loc} in $CeRhIn_5$.

Q3: Perhaps a more minor point: on page 6, when analyzing the temperature dependence of the resistivity, the authors associate the A -coefficient of the power-law dependence ($\rho = \rho_0 + AT^n$) with the effective mass of the quasiparticles. This is only true in the Fermi liquid regime $n = 2$ where $A \sim (m^*)^2$, however it is not clear what basis there is, if any, to adopt the same treatment for the non-FL regime near P_{c2} , where the quasiparticles might not even be well defined! In fact, one could have fitted the resistivity in that regime as $\rho = \rho_0 + AT^2 + BT$, and the data would suggest that the Fermi-liquid coefficient A would vanish at P_{c2} , rather than diverge, yielding the opposite conclusion regarding the effective mass. Can the authors explain why such analysis cannot be adopted?

Response: We agree that the relationship $A \sim (m^*)^2$ is only true in the Fermi liquid regime with $n = 2$. If we fix $n = 2$, the obtained pressure dependence of the coefficient A increases as pressure approaches P_{c2} from the paramagnetic regime, as shown in Fig. R1-8(d), which is similar to the coefficient A obtained from the power-law fitting. For consistency, all the parameters in Fig. 4(b–

c) are obtained from the power-law fitting in our manuscript. The coefficient A from the power-law fitting may not be directly related with the effective mass with the relationship $A \sim (m^*)^2$, but it still reflects the information related with effective mass, as reported in Ref. 19.

Fig. R1-8 (a–c) Low-temperature resistivity is plotted against T^2 at representative pressures above P_{c2} . Red arrows mark the Fermi-liquid temperature T_{FL} below which $\rho = \rho_0 + AT^2$. (d) Pressure dependence of the coefficient A (left-axis, blue circles) and the Fermi-liquid temperature T_{FL} (right-axis, red circles) defined by the temperature where the resistivity deviates from a T^2 dependence.

As for the $\rho = \rho_0 + AT^2 + BT$ analysis, the 2nd and 3rd term corresponds to the T^2 Fermi-liquid and the T -linear non-Fermi-liquid behavior contributions. As pressure approaches to the non-Fermi-liquid regime near P_{c2} , the temperature range with $\rho = \rho_0 + AT^2$ decreases, as shown by the Fermi-liquid temperature T_{FL} in Fig. R1-8. Near P_{c2} , the non-Fermi-liquid behavior with a T -linear temperature dependence dominates, and T_{FL} almost goes to zero temperature. Therefore, this fitting analysis does not seem appropriate to obtain information of the Fermi-liquid A coefficient.

Per recommendation, we have added Fig. R1-8 in the supplementary materials (Fig. S8) to show the pressure dependence of the Fermi-liquid A coefficient in the Fermi-liquid regime.

Reply to Referee #2

The manuscript deals with charge carrier delocalization transitions across (magnetic) quantum critical points, that appear to be relevant to many classes of strongly correlated electron systems and to be essential for understanding high-temperature superconductivity. The material under investigation is the well-studied heavy fermion compound CeRhIn₅. A previous de Haas-van Alphen (dHvA) study (Ref. 15 of the manuscript) indicated that in this material a drastic change of the Fermi surface occurs at a critical pressure of 2.3 GPa. A jump in the dHvA frequency and a substantial mass enhancement towards the critical pressure were extracted using magnetic fields between 10 and 16.9 T.

The present work uses Hall effect measurements to scrutinize these results. The measurements are done in a much smaller magnetic field (1 T) which has the benefit that the material is much closer to the $B = 0$ quantum critical point. The claim is that the Hall effect measurements allow identifying “an energy scale E_{loc} that signals a local-to-itinerant crossover of $4f$ degrees-of-freedom and terminates at the antiferromagnetic QCP”, thereby “establishing the origin of the Fermi surface reconstruction observed in the quantum oscillation measurements and the Kondo-breakdown nature of quantum criticality”.

Q1: I am not convinced that the presented data allow drawing such strong conclusions. The experimental signatures that the authors use are minima in the temperature-dependent Hall coefficient $R_{\text{H}}(T)$, measured at different fixed pressures across the temperature–pressure phase diagram of CeRhIn₅ and of Sn-doped CeRhIn₅. $T^*(P)$ is the curve connecting the temperatures of the R_{H} minima below a critical pressure, $T_{\text{L}}(P)$ above it.

As sketched in Fig. 1, the Fermi surface is expected to differ on the two sides of E_{loc} . Ideally, this is probed in the $T = 0$ limit as a function of the tuning parameter δ , as the notion of the Fermi surface is sharply defined only at $T = 0$. In such an experiment, the Hall coefficient should exhibit a jump at δ_{c} in the zero-temperature limit. It is not clear to me why this would translate into minima

in $R_H(T)$ curves. It is clear that with decreasing temperature the “large” Fermi surface builds up very gradually, which may or may not be reflected in a change of R_H (there are many potential sources of temperature-dependent Hall effects in multiband heavy fermion systems). But on the other side of E_{loc} , where the “small” Fermi surface exists, such a change would then not be expected, but still is observed in the experiments.

To me, it seems that the minima in $R_H(T)$ (and thus the $T^*(P)$ and $T_L(P)$ curves) are indicative of the boundaries of the quantum critical fan but cannot necessarily inform us whether or not the Hall effect will jump (as a function of P) in the zero-temperature limit.

Response: One of the hallmarks of the quantum critical fan is the strange metallic behavior characterized by a non-Fermi-liquid behavior in the resistivity. (i) In CeRhIn₅, as shown in Fig. R2-1(a), such a strange metallic state in the proximity of a quantum critical point is characterized by a sub- T linear electrical resistivity and bounded by two characteristic temperatures T_{UP}^* and T_{LO}^* . The upper temperature boundary T_{UP}^* is roughly half the resistivity maximum temperature T_{max} , which is generally associated with the onset of coherent Kondo scattering. The lower temperature boundary T_{LO}^* is a crossover where the resistivity temperature exponent n changes from 0.85 to 1. For $P < P_c$, the characteristic temperature $T^*(P)$ obtained from the minimum in $R_H(T)$ is similar to the lower temperature boundary T_{LO}^* , which is associated with the onset temperature of short-range spin–spin correlations that evolve into long-range antiferromagnetic order below T_N (Ref. 25). However, for $P > P_c$, the characteristic temperature $T_L(P)$ obtained from the minimum in $R_H(T)$ does not match the lower temperature boundary T_{LO}^* . (ii) In Sn-doped CeRhIn₅, a linear T -dependent resistivity is centered around the magnetic critical pressure P_{c2} , as shown in Fig. 4(b) in the main text. If we assume the characteristic temperatures $T^*(P)$ and $T_L(P)$ obtained from the minimum in $R_H(T)$ are indicative of the boundaries of the quantum critical fan, we should expect the convergence of $T^*(P)$ and $T_L(P)$ at the critical pressure P_{c2} in Sn-doped CeRhIn₅. However, as shown in Fig. R2-1(b), the extrapolation of $T_L(P)$ to zero temperature terminates at a lower critical pressure P_{c1} , indicating that T_L is not related to the quantum critical fan.

Fig.2-1 (a) T - P phase diagram of CeRhIn_5 . Colours represent the local exponent at zero magnetic field, adopted from Park et al., Nature 456, 366 (2008). The black squares and blue balls represent the upper temperature boundary T_{UP}^* and lower temperature boundary T_{LO}^* of the strange metallic state. The navy diamonds represent the characteristic temperature T^* , obtained from the minimum in the Hall coefficient R_{H} below the critical pressure P_{c} . The wine squares represent the onset temperature of short-range spin-spin correlations obtained from the nuclear quadrupole resonance measurements, adopted from Kawasaki et al., J. Phys. Condens. Matter 17, S889 (2005). The purple diamonds represent the characteristic temperature T_{L} , obtained from the minimum in the Hall coefficient R_{H} above the critical pressure P_{c} . Dashed lines are guides for the eyes. (b) T - P phase diagram of Sn-doped CeRhIn_5 . The navy squares and crosses represent the AFM transition determined from the first derivative of resistivity at 0 T and 4.9 T, respectively. The violet circles denote the superconducting transition temperature T_{c} determined by the zero-resistivity temperature. The purple circles and orange triangles represent the characteristic temperature T^* and T_{L} , obtained from the minimum in the Hall coefficient R_{H} below and above the critical pressure P_{c1} .

As the referee pointed out that “with decreasing temperature the “large” Fermi surface builds up very gradually, which may or may not be reflected in a change of R_{H} ”. And it was our motivation to study the temperature dependence of R_{H} to understand the relationship between R_{H} and E_{loc} . We note that, in previous studies, the energy scale of E_{loc} could be determined not only by the

isothermal thermodynamic and transport properties, but also by a change in the temperature dependent measurements.

(a) In pure and doped YbRh_2Si_2 , the E_{loc} was identified as: (i) an inflection in the isothermal field-dependent Hall coefficient and magnetoresistivity (Fig. R2-2, Ref. 13, 18); (ii) a crossover or kink in the isothermal field-dependent magnetostriction, Hall resistivity, and magnetization (Ref. 14); (iii) a maximum in the temperature dependence of susceptibility, as shown in Fig. R2-3.

Fig. R2-2 (a) Isothermal Hall coefficient of YbRh_2Si_2 as a function of field. (b) Isothermal magnetoresistivity of YbRh_2Si_2 as a function of field. Adopted from Friedemann et al., PNAS 107, 14547 (2010).

Fig. R2-3 Temperature-dependent susceptibility of $\text{Yb}(\text{Rh}_{0.94}\text{Ir}_{0.06})_2\text{Si}_2$ (a) and $\text{Yb}(\text{Rh}_{0.93}\text{Co}_{0.07})_2\text{Si}_2$ (b) at selected magnetic fields. The red arrows indicate the temperatures of the maxima assigned to the delocalization energy scale. Adopted from Friedemann et al., Nat. Phys 5, 465 (2009).

(b) In CeIn_3 , E_{loc} was assigned as: (i) a characteristic temperature at which the nuclear quadrupole resonance nuclear-spin-lattice relaxation rate $1/T_1$ deviates from its constant value (Fig. R2-4(a)); (ii) a sudden change in the nuclear quadrupole resonance frequency $\nu_Q(P)$ (Fig. R2-4(b)); (iii) a minimum in the temperature dependence of the Hall coefficient $R_H(T)$ (Fig. R2-5(a)). The delocalization crossover temperature E_{loc} or T_L chosen by these measurements are consistent, as represented by the purple and violet triangles in Fig. R2-5(b). The termination of E_{loc} within the magnetic dome is consistent with the expectation for the SDW quantum criticality in CeIn_3 .

Fig. R2-4 (a) Temperature dependence of the nuclear-spin-lattice relaxation rate $1/T_1$ for CeIn_3 at representative pressures. The dotted line indicates a relation of $1/T_1 = \text{const.}$ Dotted and solid arrows point to the AFM transition T_N and the delocalization crossover temperature T_L , respectively. (b) Pressure dependence of the nuclear quadrupole resonance frequency ν_Q at 10 K. ν_Q begins to increase significantly above 2 GPa due to the delocalization. Adopted from Kawasaki et al., *Phy. Rev. B* 77, 064508 (2008).

Fig. R2-5 (a) Temperature dependence of the Hall coefficient $|R_H|$ for CeIn_3 under high pressure. (b) T - P phase diagram of CeIn_3 . The red and blue squares indicate the AFM transition T_N and

superconducting transition T_c , respectively. The violet and purple triangles indicate the delocalization crossover T_L , determined by the Hall minimum and the deviation temperature in the nuclear quadrupole resonance measurements, respectively. Inset shows the pressure dependence of the resistivity exponent n . Adopted from Araki et al., J. Phys. Soc. Jpn, 84, 123702 (2015) and Knebel et al., Phys. Rev. B 65, 024425 (2001).

In CeIn_3 , as shown in the inset of Fig. R2-5(b), a quantum critical fan is centered around the critical pressure $P_c \sim 2.5$ GPa, where the AFM transition is suppressed to 0 K and the resistivity shows a non-Fermi-liquid behavior (Knebel et al., Phys. Rev. B 65, 024425 (2001)). However, the boundary obtained from the minimum in $R_H(T)$ does not terminate at P_c , contrary to the assumption that “*the minima in $R_H(T)$ are indicative of the boundaries of the quantum critical fan*”. Instead, the fact that the Hall minimum temperature is consistent with the characteristic temperature at which the nuclear-spin-lattice relaxation rate $1/T_1$ deviates from its constant value in the nuclear quadrupole resonance measurements indicates that the characteristic temperature obtained by the minima in $R_H(T)$ is associated with the delocalization of magnetic moments.

Considering the similarities between CeIn_3 and CeRhIn_5 the minimum in the temperature dependence of Hall coefficient $R_H(T)$ in CeRhIn_5 can also be a signature for E_{loc} . Especially, the Hall minimum temperature in CeRhIn_5 extrapolated from the paramagnetic side terminates at the magnetic QCP where a sudden change of Fermi surface from small to large has been observed by the de Haas-van Alphen measurements, which further supports the correspondence between the Hall minimum temperature and the charge delocalization crossover energy scale of E_{loc} above P_c . The nuclear quadrupole resonance measurements can be used to further confirm this the energy scale E_{loc} in pure and Sn-doped CeRhIn_5 .

In the low-pressure AFM regime, on the other hand, the Hall minimum temperature decreases with increasing pressure and terminates in the zero-temperature limit at the magnetic QCP in pure and Sn-doped CeRhIn_5 . The Hall minimum temperature is consistent with the onset temperature of short-range spin-spin correlations in CeRhIn_5 observed in the nuclear quadrupole resonance measurements (Ref. 25, 26). These results imply that, although a Hall minimum also appears in the small Fermi surface regime, it has a different origin.

Per recommendation, we have added the discussion of E_{loc} in $CeIn_3$ in the second paragraph of section “Observation of E_{loc} in $CeRhIn_5$ ” on page.5 as another supporting evidence for the relationship between the Hall minimum temperature and E_{loc} in $CeRhIn_5$. In the abstract part, we have changed “*Herein, the ambiguity of the nature of quantum criticality in the prototypical quantum critical superconductor $CeRhIn_5$ is removed by identifying an energy scale E_{loc} that signals a local-to-itinerant crossover of $4f$ degrees-of-freedom and terminates at the antiferromagnetic QCP, establishing the origin of the Fermi surface reconstruction observed in the quantum oscillation measurements and the Kondo-breakdown nature of quantum criticality. The termination of E_{loc} inside the magnetically ordered phase in Sn-doped $CeRhIn_5$ implies the evolution of quantum criticality from the Kondo-breakdown to spin-density-wave type with Sn substitution.*” to “Herein, we use pressure-dependent Hall measurements to identify a finite-temperature scale E_{loc} that signals a crossover from f -localized to f -delocalized character. As a function of pressure, $E_{loc}(P)$ extrapolates smoothly to zero temperature at the antiferromagnetic QCP of $CeRhIn_5$ where its Fermi surface reconstructs, hallmarks of Kondo-breakdown criticality that generates critical magnetic and charge fluctuations. In 4.4% Sn-doped $CeRhIn_5$, however, $E_{loc}(P)$ extrapolates into its magnetically ordered phase and is decoupled from the pressure-induced magnetic QCP, which implies a spin-density-wave (SDW) type of criticality that produces only critical fluctuations of the SDW order parameter.”.

Q2: I would like to encourage the authors to plot (from their $R_H(T)$ data at fixed P) $R_H(P)$ curves at a set of different fixed temperatures. Is a Hall crossover seen? If yes, at which pressure? And does it sharpen with decreasing temperature to be indicative of a jump in the $T = 0$ limit?

Response: Per recommendation, we have added Fig. R2-6 in the supplementary materials (Fig. S9) to show the isothermal pressure dependent Hall coefficient in pure and Sn-doped $CeRhIn_5$.

Unlike $YbRh_2Si_2$ where a step-like crossover appears under magnetic field, a step-like crossover was not observed in the isothermal pressure dependence of Hall coefficient in $CeRhIn_5$ even though dHvA measurements revealed the Fermi surface jump at the QCP. The isothermal Hall coefficient $|R_H(P)|$, as shown in Fig. R2-6(a), increases rapidly as the magnetic QCP P_c is approached from either the magnetic or paramagnetic regimes, which is consistent with previous

Hall report (Ref. 24). A similar pressure dependence of $|R_H|$ is also observed in Sn-doped CeRhIn_5 , as shown in Fig. R2-6(b), but the maximum of $|R_H|$ appears deep in the magnetic state which is located at a lower pressure than the magnetic QCP.

Fig. R2-6 (a) Pressure dependence of Hall coefficient $|R_H(P)|$ for CeRhIn_5 at 2.4 K. P_c indicates the magnetic QCP. (b) Pressure dependence of Hall coefficient $|R_H(P)|$ for Sn-doped CeRhIn_5 at representative temperatures. P_{c2} denotes the magnetic QCP. P_{c1} indicates the critical pressure where T_L extrapolates to zero temperature and is lower than the magnetic QCP P_{c2} .

The deviation of the pressure dependence of $|R_H|$ in pure and Sn-doped CeRhIn_5 from the ideal case arises because Hall effects in multiband heavy fermion systems could be affected by different sources. In a system like CeRhIn_5 , it can be influenced not only by changes in carrier density but also by scattering rates on different parts of the Fermi surface, details of the surface topology, and the presence of spin fluctuations. We note that the appearance of AFM transition can also affect the isothermal Hall effects. For example, in $\text{Ce}_3\text{Pd}_{20}\text{Si}_6$, there are two crossover behavior below the AFM transition: as shown in Fig. R2-7(a), one is related with the charge delocalization as shown in YbRh_2Si_2 , while the other is related with the AFM transition, which is not observed in YbRh_2Si_2 . Similarly, the isothermal field dependence of magnetoresistivity below the AFM transition contains a background contribution, a crossover due to the delocalization, and an increase of resistivity at low magnetic fields due to the AFM transition (Fig. R2-7(b-c)). In this kind of compounds, other contributions need to be removed to get a crossover due to the delocalization, as shown in R2-7(c).

Fig. R2-7 (a) Isothermal Hall resistivity of $\text{Ce}_3\text{Pd}_{20}\text{Si}_6$ as a function of magnetic field at different temperatures below the AFM transition. (b) Isothermal magnetoresistivity of $\text{Ce}_3\text{Pd}_{20}\text{Si}_6$ as a function of magnetic field. (c) Representative fitting procedure for magnetoresistivity corrected for the background contribution ($\rho_l - \rho_{l,\text{back}}$). B_N and B^* represent two crossover fields. The blue dashed and red solid lines represent the contribution from the AFM transition and delocalization, respectively. Adopted from Custers et al., Nat. Mater 11, 189 (2012).

Q3: Another point is whether the authors have firm evidence that the AFM phase transition remains continuous up to P_c at the field of $B = 1 \text{ T}$ (used in their R_H measurements).

Response: Since the transport measurements cannot detect the AFM transition inside the superconducting phase, the external magnetic field is used to suppress the superconducting phase and to investigate the dependence on pressure of T_N near the projected AFM QCP under the superconducting dome. Figure R2-8 shows the T - P phase diagram of Sn-doped CeRhIn_5 , where T_{NS} at 0 T and 4.9 T are represented by navy squares and crosses, respectively. As shown in the figure, T_{NS} are almost independent of the magnetic field and observed even at 1.2 GPa, which is much larger than P_{c1} of 1.0 GPa. These results indicate that T_{NS} are smoothly extended to P_{c2} in Sn-doped CeRhIn_5 . Furthermore, the observation of both T_N and T_L at 1.2 GPa shows that both characteristic temperatures remain finite in the critical regime between P_{c1} and P_{c2} , supporting that they are decoupled and terminate at two different pressures. As shown in the pure CeRhIn_5 (Ref. 21, 22), however, other measurements such as heat capacity that can detect the phase transition in the superconducting phase are required to make a definitive conclusion

Fig. R2-8. T - P phase diagram of Sn-doped CeRhIn_5 . The navy squares and crosses represent the AFM transition determined from the first derivative of resistivity at 0 T and 4.9 T, respectively. The violet circles denote the superconducting transition temperature T_c determined by the zero-resistivity temperature. The purple circles and orange triangles represent the onset of short-range magnetic correlations at T^* and the $4f$ -electron delocalization crossover temperature T_L , obtained by the minimum of Hall coefficient $R_H(T)$ at 5 T.

Q4: I'd be happy to be convinced that the author's (very interesting!) conclusions are robust, but do judge the evidence put forward in the present manuscript insufficient. So, unfortunately, I cannot recommend publication in the present form.

Response: We believe that our additional measurements near the critical pressures better support the conclusion of the decoupled two critical pressures in Sn-doped CeRhIn_5 , where T_L and T_N were observed between P_{c1} and P_{c2} . However, more conclusive measurements such as heat capacity and Fermi surface measurement are required to make an even more definitive conclusion. To reflect this, we change the title from "Charge delocalization crossover in the quantum critical superconductor CeRhIn_5 " to "Evidence for charge delocalization crossover in the quantum critical superconductor CeRhIn_5 ".

Reply to Referee #3

Wang et al. present Hall effect and resistance measurements on CeRhIn₅ and Sn-doped CeRhIn₅ and claim that these demonstrate the separation of a local energy scale from the magnetic quantum critical point. The results are of interest to a large community of researchers working on quantum criticality and underpins research in the wider area of correlated materials. I recommend revision of this manuscript before it is accepted for publication. Whilst the experimental data are of excellent quality, the interpretation of the results should be clarified.

Q1: The authors associate extrema in the temperature dependence of the Hall coefficient with the magnetic fluctuations at low pressures and with a Fermi surface reconstruction at high pressures. Arguments are provided that the high-pressure extrema are not associated with magnetic fluctuations based on the insensitivity to magnetic field. However, it would be good to also discuss and show evidence for the low-pressure extrema to be associated with spin fluctuations, i.e. T^* . Is there a larger sensitivity to magnetic field for the low-pressure extrema? In fact, it appears that ref 24 has observed a large field dependence of the Hall coefficient at 2.0 GPa. Has this been reproduced by the present study?

Response: In pure and Sn-doped CeRhIn₅, the characteristic temperature $T^*(P)$ obtained by the minimum in Hall coefficient $R_H(T)$ is higher than the long-range AFM transition temperature T_N , decreases with pressure, and finally terminates in the zero-temperature limit at the magnetic QCP, at which the AFM transition is also suppressed to zero temperature. The Hall minimum temperature $T^*(P)$ is similar to the onset temperature of short-range spin–spin correlations observed in CeRhIn₅ by the nuclear quadrupole resonance measurements (Ref. 25, 26), which evolve into long-range antiferromagnetic order below T_N . Therefore, $T^*(P)$ could be associated with the onset of short-range spin correlations in pure and Sn-doped CeRhIn₅ at pressures below the AFM QCP.

The temperature dependence of Hall coefficient at different fields is plotted in Fig. R3-1. Although

the amplitude of R_H is affected by the magnetic field, the position of T^* is almost insensitive to the magnetic field in both pure and Sn-doped CeRhIn₅.

Fig. R3-1 Temperature dependence of Hall coefficient at different fields for CeRhIn₅ at ambient pressure (a) adopted from Hundley et al., Phys. Rev. B 70, 035113 (2004) and Sn-doped CeRhIn₅ at 0.03 GPa and 0.48 GPa (b–c).

The temperature dependence of the Hall voltage $V_H(B) = [V_H(B) - V_H(-B)]/2$ was measured under the fields of 12 T and 1 T at $P = 2.4 \text{ GPa}$ in CeRhIn₅. The Hall coefficient was calculated from the standard expression $R_H = V_H/IB$, where I is the applied current flowing perpendicular to the applied field B , and t is the sample thickness. As shown in Fig.R3-2, R_H is much suppressed by the applied field, implying a large field dependence of the Hall coefficient at 2.4 GPa. These results suggest a large field dependence of the Hall coefficient at 2.0 GPa in CeRhIn₅.

Fig. R3-2 Temperature dependence of Hall coefficient for CeRhIn₅ at 2.4 GPa measured at different fields.

The magnetic field dependences of Hall resistivity of Sn-doped CeRhIn₅ are shown at several representative pressures in Fig.R3-3. The almost linear field dependence of Hall resistivity in the Sn-doped CeRhIn₅ indicates that a weak field dependence of Hall coefficient in the observed pressure range. It can also be seen from the Supplementary Fig.S3.

Fig. R3-3 Field dependence of Hall resistivity for Sn-doped CeRhIn₅ at representative pressures measured at 0.3 K.

Per recommendation, we have added the discussion of field effect on T^* in the first paragraph of section “Observation of E_{loc} in CeRhIn₅” on page.4 and the Fig. R3-3 in the supplementary materials to show the weak field dependence of Hall coefficient in Sn-doped CeRhIn₅.

Q2: Can the authors comment on the isothermal pressure dependence of the Hall coefficient. It appears from their contour plots that this will display a single maximum. How can this be reconciled with two separate energy scales?

Response: As shown in the contour plots and Fig. R3-4, the isothermal pressure dependence of

the Hall coefficient displays a single maximum in pure and Sn-doped CeRhIn₅. The difference between these two compounds is where the Hall maximum appears. In CeRhIn₅ (Fig. R3-4(a)), the isothermal Hall coefficient $|R_H(P)|$ increases rapidly as the magnetic QCP P_c is approached from either the magnetic or paramagnetic regimes, and the Hall maximum appears at the magnetic QCP, at which all characteristic temperatures, including the AFM transition T_N , the onset temperature of short-range spin correlations T^* , and the delocalization crossover temperature T_L , converge. On the other hand, although a similar pressure dependence of the Hall coefficient is also observed in Sn-doped CeRhIn₅ at low temperatures (Fig. R3-4(b)), the Hall maximum appears at a pressure below the magnetic QCP, at which only the delocalization crossover temperature T_L terminates to zero temperature. Therefore, the extremum in $|R_H(P)|$ appears to be related to the consequences of Kondo breakdown in pure and Sn-doped CeRhIn₅.

Fig. R3-4 (a) Pressure dependence of Hall coefficient $|R_H(P)|$ of CeRhIn₅ at 2.4 K. P_c indicates the magnetic QCP. (b) Pressure dependence of Hall coefficient $|R_H(P)|$ of CeRh(In_{0.956}Sn_{0.044})₅ at 0.3 K (solid red balls, right y-axis) along with the T - P phase diagram (left y-axis). The orange circles represent the AFM transition T_N determined from the first derivative of resistivity at 0 T. The violet circles denote the superconducting transition temperature T_c determined by the zero-resistivity temperature. The purple squares and orange triangles represent the onset of short-range magnetic correlations at T^* and the $4f$ -electron delocalization crossover temperature T_L , obtained by the minimum of Hall coefficient $R_H(T)$ at 5 T. P_{c1} is the critical pressure where the amplitude of R_H reaches a maximum and T_L terminates to 0 K. P_{c2} is the AFM QCP where T_N and T^* extrapolate to 0 K and T_c is maximum. Dashed red lines are guides to the eyes.

The characteristic temperature $T^*(P)$ in the low-pressure regime is associated with the onset temperature of short-range spin–spin correlations that evolve into the long-range antiferromagnetic order below T_N (Ref. 25, 26). With increasing pressure, T^* and T_N are suppressed together and terminate in the zero-temperature limit at the magnetic QCP in pure and Sn-doped CeRhIn₅. The magnetic fluctuations are induced with the suppression of magnetic order, which is not shown as a maximum in the isothermal pressure dependence of the Hall coefficient. The effect of magnetic fluctuations is manifested as a non-Fermi liquid behavior in the resistivity behavior near the magnetic QCP, as shown in Fig.4 in the main text.

REVIEWER COMMENTS

Reviewer #1 (Remarks to the Author):

It is refreshing to see that all three Referees have raised the same two questions to the authors -- (i) why should one believe their association of Hall effect minimum T_L with the localization transition and (ii) can the authors plot the Hall isotherms as a function of pressure $R_H(P)$, and will they see any kinks or sharpening similar to what has been reported in YRS?

While the authors have presented the arguments based on identification of T_L in the related compound $CeIn_3$ to support their identification regarding point (i), I am disappointed that none of the discussion made it into the main text. A single sentence introduced on page 5 is too short and cryptic to do it justice. I would strongly encourage the authors to transfer a (perhaps abbreviated) version of the response they gave to the Referees to the main text.

Even more disappointing is the complete lack of discussion of item (ii), the Hall isotherms (which all the Referees requested) in the main text. I am a believer in that the published paper should stand on its own, even without considering the evidence presented in the SM. I am therefore not satisfied with the only discussion of this crucially important point to be relegated to the SM. I would insist that the authors make the figure S9 (or the figure R3-4 in response to the Third Referee) to be shown explicitly in the main text, with the accompanying discussion. This is very important, especially for the readers familiar with the Hall measurements in YRS, to appreciate the subtleties and differences with $CeRhIn_5:Sn$ discussed by the authors in their response to the Referees.

I would only recommend the manuscript for publication provided the above two points are explicitly discussed in the main text.

Reviewer #2 (Remarks to the Author):

I am satisfied with the replies of the authors and with their changes to the manuscript and supplementary information and recommend publication of the work in the present form.

Reviewer #3 (Remarks to the Author):

The authors have provided figures that demonstrate the pressure dependence of the Hall coefficient. These figures show convincingly that the maximum in $RH(P)$ is located at the critical pressure of the AFM state for $CeRhIn_5$ whilst the maximum in $RH(P)$ is located at a pressure P_{c1} below the AFM for $CeRh(In_{0.956}Sn_{0.044})_5$. Unfortunately, the discussion why a single maximum is consistent with two energy scales remains vague. The authors state that in “the Hall effect in multiband heavy-fermion systems can be influenced not only by changes of the carrier density but also by scattering rates on different parts of the Fermi surface, details of the surface topology, and the presence of spin fluctuations.” The authors do not provide clear arguments which of these is the dominant contribution for the maximum in $RH(P)$. Only the change in carrier density can be linked to the Fermi surface reconstruction associated with the claimed localisation energy scale. Hence, I remain sceptical as the claim of the manuscript is not supported by the data. I cannot recommend publication of the manuscript.

We would like to thank the referees for the suggestions and comments on our manuscript. Below are our point by point replies to the comments. Changes in the main text and SI have been highlighted in red font.

Reply to Referee #1

It is refreshing to see that all three Referees have raised the same two questions to the authors -- (i) why should one believe their association of Hall effect minimum T_L with the localization transition and (ii) can the authors plot the Hall isotherms as a function of pressure $R_H(P)$, and will they see any kinks or sharpening similar to what has been reported in YRS?

Q1: While the authors have presented the arguments based on identification of T_L in the related compound $CeIn_3$ to support their identification regarding point (i), I am disappointed that none of the discussion made it into the main text. A single sentence introduced on page 5 is too short and cryptic to do it justice. I would strongly encourage the authors to transfer a (perhaps abbreviated) version of the response they gave to the Referees to the main text.

Response: Per recommendation, we have modified the discussion of E_{loc} in $CeIn_3$ in the second paragraph of section “Observation of E_{loc} in $CeRhIn_5$ ” on page 5 to support the relationship between the Hall minimum temperature and E_{loc} in $CeRhIn_5$ and added the abbreviated version of the response as the first paragraph in the section “Discussion” on page 6 to give more detail discussion of signatures of E_{loc} in different samples and measurements.

Q2: Even more disappointing is the complete lack of discussion of item (ii), the Hall isotherms (which all the Referees requested) in the main text. I am a believer in that the published paper should stand on its own, even without considering the evidence presented in the SM. I am therefore not satisfied with the only discussion of this crucially important point to be relegated to the SM. I would insist that the authors make the figure S9 (or the figure R3-4 in response to the Third Referee) to be shown explicitly in the main text, with the accompanying discussion. This is very important, especially for the readers familiar with the Hall measurements in YRS, to appreciate the subtleties and differences with $CeRhIn_5:Sn$ discussed by the authors in their response to the Referees.

Response: Per recommendation, we have included Fig.S9 in the main text as Figure 5 and the relevant discussion in the section “Discussion” on page 7.

I would only recommend the manuscript for publication provided the above two points are explicitly discussed in the main text.

Response: We thank the Referee for the suggestions that will be very helpful to better understand this work. After addressing the comments, we believe that the manuscript is better suited for publication in Nature Communications.

Reply to Referee #2

I am satisfied with the replies of the authors and with their changes to the manuscript and supplementary information and recommend publication of the work in the present form.

Response: We are grateful to Referee #2 for recommending publication of our manuscript.

Reply to Referee #3

The authors have provided figures that demonstrate the pressure dependence of the Hall coefficient. These figures show convincingly that the maximum in $R_H(P)$ is located at the critical pressure of the AFM state for CeRhIn_5 whilst the maximum in $R_H(P)$ is located at a pressure P_{c1} below the AFM for $\text{CeRh}(\text{In}_{0.956}\text{Sn}_{0.044})_5$. Unfortunately, the discussion why a single maximum is consistent with two energy scales remains vague. The authors state that in “the Hall effect in multiband heavy-fermion systems can be influenced not only by changes of the carrier density but also by scattering rates on different parts of the Fermi surface, details of the surface topology, and the presence of spin fluctuations.” The authors do not provide clear arguments which of these is the dominant contribution for the maximum in $R_H(P)$. Only the change in carrier density can be linked to the Fermi surface reconstruction associated with the claimed localization energy scale. Hence, I remain skeptical as the claim of the manuscript is not supported by the data. I cannot recommend publication of the manuscript.

Response: We thank the referee for making these important points that we address below. The first point concerns, we believe, why there is a single maximum in $|R_H(P)|$ even though there are two critical points in $\text{CeRh}(\text{In}_{0.956}\text{Sn}_{0.044})_5$. As the referee notes and as discussed in the text, the Hall coefficient depends not only on the carrier density but also on several other factors, especially in the vicinity of a QCP. In CeRhIn_5 , the magnetic QCP coincides with a change of Fermi surface and there is a single sharp maximum in $|R_H(P)|$. But in Sn-doped CeRhIn_5 , two critical points at P_{c1} and P_{c2} are separated by a small pressure difference of less than 0.5 GPa. In this case, there is not a strong peak in $|R_H(P)|$ but a broad maximum that encompasses both critical points due to contributions from each of the closely spaced QCPs and the presence of disorder. Indeed, it is common to find a broadened response even when two classical phase transitions are nearby each other. We hope that this reply addresses the referee's concern.

The second point concerns the lack of definitive evidence for change in carrier density linked to the Fermi surface reconstruction associated with the localization energy scale. We agree completely with the referee that unambiguous evidence for a change in carrier density would be ideal, but even in much studied YbRh_2Si_2 there is no claim, to our knowledge, of direct evidence for a jump in carrier density but only for a jump in Hall coefficient that is 'inferred' to reflect a change in Fermi volume. On general grounds, the Hall coefficient should be influenced by critical charge fluctuations associated with a $T = 0$ K localized-delocalized transition and these will complicate if not mask a direct measurement of the carrier density. Because of this, we must respectfully disagree with the referee's statement that 'Only the change in carrier density can be linked to the Fermi surface reconstruction associated with the claimed localization energy scale.'

As stated in the first paragraph and throughout the text of our manuscript, we present evidence that a localization-delocalization crossover scale can be identified from Hall measurements and then place that evidence in the context of direct evidence for Fermi surface reconstruction from dHvA studies in CeRhIn_5 . We do not claim that our Hall measurements directly measure a carrier density, and it would be inappropriate to do so. We do, however, claim that identifying a localization-delocalization scale with a minimum in $R_H(T)$ provides a straightforward interpretation within the context of a Kondo-breakdown scenario and that magnetic and charge fluctuations accompanying localization-delocalization manifest in the Hall coefficient.

In order to clarify these points, we have included the isothermal pressure dependence of the Hall coefficient in the main text as Fig. 5 and added discussion of other quantum critical systems where a localization-delocalization scale has been found in the discussion section.